

# Temporal patterns of greenhouse gas emissions from two small thermokarst lakes in Nunavik, Canada

Amélie Pouliot[1,3]; Isabelle Laurion[2,3]; Antoine Thiboult[1]; Daniel F. Nadeau[1]

[1]Department of Civil and Water Engineering, Université Laval, Québec, Qc
[2]Centre Eau Terre Environnement, Institut national de la recherche scientifique, Québec, Qc
[3]Centre d'études nordiques, Université Laval, Québec, Qc

*Correspondence*: Amélie Pouliot (amelie.pouliot.1@ulaval.ca) and Daniel Nadeau (daniel.nadeau@gci.ulaval.ca)

**Abstract.** Small thermokarst lakes, formed by the thawing of ice-rich permafrost, are significant sources of greenhouse gases (GHG). Most estimates of emissions rely solely on daily measurements, which may bias annual flux calculations. In this study, we combined GHG flux measurements from two intensive summer campaigns with nearly two years of continuous temperature, oxygen, and conductivity profiling in two small (<200 $m^2$) thermokarst lakes in Nunavik (56°33'28.8"N, 76°28'46.5"W), Canada. One campaign occurred during a colder summer (0.7°C above the seasonal mean) and the other during a warmer one (2.6°C above the seasonal mean), with one lake being humic and sheltered and the other more transparent and wind-exposed. Average diffusive fluxes of $CO_2$ (22.1 ± 20.5 mmol $m^{-2}$ $d^{-1}$; mean ± standard deviation) and $CH_4$ (14.3 ± 14.2 mmol $CO_2$-eq $m^{-2}$ $d^{-1}$) were consistent with values reported for similar thermokarst lakes, while $N_2O$ fluxes were negligible (−0.8 ± 1.3 mmol $CO_2$-eq $m^{-2}$ $d^{-1}$). Emissions increased 4-fold during the warmer summer, alongside the emergence of a diel trend, where daytime (09:00-17:00) $CO_2$ fluxes increased by 47%, $CH_4$ by 95%, and negative $N_2O$ fluxes by 75% relative to nighttime fluxes. Moreover, ebullitive $CH_4$ fluxes were six times higher than diffusive fluxes in the humic, sheltered lake, reaching 117.0 ± 44.7 mmol $CO_2$-eq $m^{-2}$ $d^{-1}$. Seasonal flux estimates indicate that emissions peaked in fall and spring, as they were almost four times higher than those in summer. Our findings highlight the importance of including both daytime and nighttime measurements, as well as storage fluxes (emitted in spring and fall), to improve the accuracy of GHG emission estimates from thermokarst lakes.

## 1 Introduction

Permafrost thaw in ice-rich soils results in the formation of thermokarst lakes, which vary widely in size−from a few hundred square meters to over 50 square kilometres. These lakes, common across Arctic and subarctic regions of Siberia, Alaska, and Canada, differ from other northern lakes in their mixing regime, transparency, and biogeochemistry (e.g., Sepulveda-Jauregui et al., 2015; Prėskienis et al., 2021; Serikova et al., 2019). They typically contain large amounts of organic carbon leached from degrading permafrost soils (Tarnocai et al., 2009; Schuur et al., 2009), often leading to significant GHG emissions, along with pronounced seasonality in emission rates (Bouchard et al., 2015; Heslop et al., 2020; Kuhn et al., 2023; Matveev et al.,





2016; Wik et al., 2016). Due to climate change and permafrost thawing, thermokarst lakes in some regions are still expanding downward and laterally, while others are experiencing drainage through changes in hydro-connectivity (Smith et al., 2005). As the ice-free season lengthens, warmer and wetter conditions may further enhance GHG production, originating either from thawing permafrost carbon or from freshly fixed carbon, each with distinct feedback effects (Prėskienis et al., 2021).

Much research to date has focused on large thermokarst lakes in areas of continuous permafrost with Yedoma soils (Pleistocene, ice-rich, carbon-rich permafrost). Thermokarst lakes in non-Yedoma landscapes of subarctic regions are typically small, shallow, and notably turbid (Grosse et al., 2013; Laurion et al., 2010; Bouchard et al., 2015; Crevecoeur et al., 2015; Wik et al., 2016). They result from the degradation of palsas (organic-rich mounds) or lithalsas (on mineral soils). Lakes on palsas tend to be dark and rich in organic matter, while those on lithalsas often display a range of colours resulting from the

presence of suspended silts and clays, as well as varying amounts of organic matter leached from the watershed (Matveev et al., 2018). In subarctic areas, thousands of thermokarst lakes are smaller than $0.1 \text{ km}^2$ and often fall outside large-scale surface area classifications (Downing, 2010; Wik et al., 2016). Moreover, studies have reported increases in thermokarst lake area in recent decades, for example in Nunavik (northern Quebec), Canada, with a 96% increase from 1957 to 2009 (Jolivel and Allard, 2013). To better assess their contribution to the global carbon budget, both their areal extent and limnological

functioning need to be evaluated.

GHG emissions from thermokarst lakes consist of diffusive and ebullitive fluxes. Diffusive emissions are influenced by gas accumulation in the upper layers of the water column and turbulence at the lake surface. Ebullitive fluxes, on the other hand, involve bubbles released from the sediments and depend on factors such as sediment temperature (Wik et al., 2014; Kuhn et al., 2021), atmospheric pressure (Goodrich et al., 2011), dominant vegetation and lake depth (Burke et al., 2019). Significant

amounts of $CH_4$ are typically released through ebullition, with high release rates often observed in lakes with carbon-rich sediments, such as thermokarst lakes (Wang et al., 2021; Wik et al., 2016; Yang et al., 2023). Many studies have shown that ebullitive fluxes are predominant in thermokarst lakes, largely overcoming diffusive fluxes (Walter et al., 2006; Sepulveda-Jauregui et al., 2015). However, other studies have reported cases where diffusive fluxes of $CO_2$ and $CH_4$ can dominate, potentially influenced by sediment type (Matveev et al., 2016; Matveev et al., 2018). $N_2O$ emissions from thermokarst lakes

are not well documented. $N_2O$ concentrations reaching up to 222% saturation have been measured in the water column of a High Arctic thermokarst lake, highlighting the need for better quantification of $N_2O$ emissions from these systems (Begin et al., 2021). However, studies have reported near-equilibrium $N_2O$ concentrations in shallow northern lakes, including boreal (Huttunen et al., 2002), thermokarst (Abnizova et al., 2012), and eutrophic systems (Davidson et al., 2024).

In dimictic lakes, gases typically accumulate under the ice cover in winter and at the bottom of stratified lakes in summer,

sometimes reaching concentrations that are orders of magnitude higher than at the surface. Therefore, spring and autumn are often marked by peaks in diffusive emissions as the water column mixes, a process known as the storage flux (Sepulveda-Jauregui et al., 2015; Greene et al., 2014). However, the mixing dynamics of thermokarst lakes differ from those of larger lakes due to their smaller fetch, partial wind sheltering, and higher concentrations of dissolved organic matter (DOM) and suspended solids. These factors can lead to distinct patterns of GHG outgassing accumulated over winter and summer,



especially during spring turnover, which may be brief or even absent in humic lakes (Matveev et al., 2019). Significant seasonal variations in diffusive emissions have been documented in thermokarst lakes across various regions (Hughes-Allen et al., 2021; Prėskienis et al., 2021; Zabelina et al., 2020).

Diel variations in diffusive emissions are also commonly observed in lakes during the open water period. While some studies report higher fluxes at night or in the early morning (Podgrajsek et al., 2014; Eugster et al., 2022; Zhao et al., 2024), others
find increased emissions during the day (Sieczko et al., 2020; Martinez-Cruz et al., 2020; Erkkilä et al., 2018). It is generally accepted that diel wind patterns are closely linked to diel flux patterns in larger lakes, where nighttime is often characterised by low wind speeds and a stable atmospheric boundary layer. However, in smaller, sheltered systems, recent research suggests that nighttime diffusive fluxes may dominate (Walter Anthony and Macintyre, 2016). This is attributed to surface cooling, which generates large eddies that bring GHG-enriched water to the surface, as well as small eddies that enhance gas exchange
at the water-air interface. Because thermokarst lakes typically contain high concentrations of chromophoric DOM (CDOM), they tend to be strongly stratified during summer, with a shallow, diurnal upper mixed layer. Thus, convection-enhanced diffusion may be a key driver of GHG emissions in these systems (Macintyre et al., 2010; Holgerson et al., 2016). Northern studies are often constrained by logistical challenges, with limited flux measurement points and a tendency to collect data only during the day–potentially introducing significant bias (Prėskienis et al., 2021; Hughes-Allen et al., 2021; Yang et al., 2023;
Kuhn et al., 2023). Diffusive fluxes are commonly estimated using wind-driven models (Cole and Caraco, 1998; Wanninkhof, 2014), but these models are less accurate for very small, wind-sheltered lakes. While the eddy covariance method provides continuous measurements, it is less effective for small lakes due to footprint area constraints. Chamber-based methods, though more suitable, are limited to point measurements.

Upscaling emissions from small thermokarst lakes remains challenging due to their dynamic nature, variable morphology, and
dependence on regional factors such as permafrost conditions, soil properties, and local meteorology. Simplified estimation methods can introduce substantial variability in results. Therefore, we highlight the need for more studies quantifying fluxes from these very small–but abundant–lakes, with careful consideration of diel and seasonal variability. Such efforts are essential for refining $CO_2$, $CH_4$, and $N_2O$ flux estimates, which are key components of the global GHG budget.

This study quantified GHG emissions from two lithalsa thermokarst lakes in a subarctic region of Canada. Despite being
located on the same degrading permafrost mound, the lakes differ in key characteristics: one is more sheltered and humic, while the other is wind-exposed and more transparent. The objectives were to (1) characterise their physicochemical properties, (2) assess the magnitude and diel variation of summer GHG fluxes, and (3) estimate seasonal variability. We conducted two contrasting summer campaigns in 2022 and 2023–one during a colder period and the other during a warmer one–focusing on temporal patterns in GHG diffusive fluxes and the magnitude of ebullitive fluxes in 2023. Additionally, we continuously
monitored water column conditions for two years, alongside hourly meteorological data, enabling extrapolation beyond the flux observation period.



## 2 Methods

### 2.1 Study site

The study site (56°33'28.8"N, 76°28'46.5"W) is located in the Tasiapik Valley, near the Inuit community of Umiujaq in
Nunavik, Quebec, Canada. In the lower part of the valley, the surface deposits are predominantly marine silt, while the upper part consists of a thin layer of littoral sands. Discontinuous permafrost started penetrating the silt unit in the lower valley around 6500 years BP. In the upper valley, the sand layer generally prevented permafrost aggradation, but in areas with a thinner sand layer, the underlying silt froze, leading to the formation of permafrost mounds (Dagenais et al., 2020; Fortier et al., 2020).

The region has a subpolar continental climate, characterised by long winters and short, cool summers. Over the period 1997-2023, the mean annual air temperature was approximately –3°C. Vegetation in the Tasiapik Valley is highly variable: the upper valley is dominated by lichen-covered mounds surrounded by shrubs, while the lower valley is primarily covered by forested tundra. This diversity of vegetation types highlights the valley's role as an ecotone, a transitional zone between tundra and boreal forest ecosystems. Climate change trends observed in Nunavik indicate that permafrost in the region has been degrading
over the past 25 years (Lemieux et al., 2016).

### 2.2 Lake surveys

We closely monitored two lakes located in the upper valley (Fig. 1), referred to as TAS1 and TAS3 for consistency with previously sampled lakes in the region (unpublished results). Two intensive summer campaigns were conducted: one from July 7-16, 2022, and another from August 9-21, 2023. Additionally, the lakes were continuously monitored from October 2021
to August 2023 to study their oxythermal structure. The two shallow thermokarst lakes are situated on a single degrading lithalsa mound, as shown in the bathymetric map (Fig. S1). Currently, they remain hydrologically isolated due to an underlying silt layer, with their interannual water balance entirely dependent on the ratio of annual precipitation to evaporation (Bussière et al., 2022). Given their shallow depth, water levels fluctuate significantly. In the future, these lakes may merge into a single waterbody. Like other lakes in the region, they are small, shallow, and strongly absorb light. Despite their size, they are
stratified, with anoxic conditions in bottom waters, similar to other subarctic thermokarst lakes (Matveev et al., 2018).

TAS1 has a surface area of 186 m$^2$ (Table 1) and is located on the south side of the permafrost mound, surrounded by shrubs, though it remains relatively exposed to the wind (Fig. 1). In July 2022, its maximum water level was $2.60 \pm 0.02$ m, slightly decreasing to $2.50 \pm 0.02$ m in August 2023. Compared to TAS3, TAS1 has lower concentrations of dissolved organic carbon (DOC), CDOM absorption (at 320 nm, $a_{320}$), and total suspended solids (TSS), making it a more transparent lake with a Secchi
depth ($Z_{SD}$) of 1.23 m. Nutrient concentrations were also very high, with total phosphorus (TP) averaging 245 µg L$^{-1}$ and total nitrogen (TN) 1337 µg L$^{-1}$, indicating high potential productivity (Table 2).



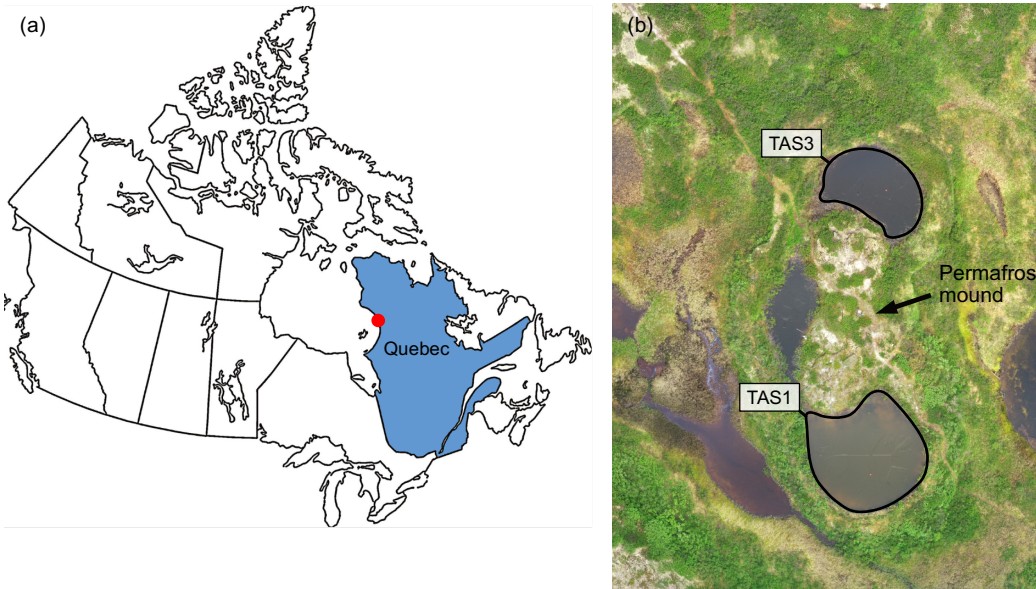

**Figure 1. Location map of the Tasiapik Valley: (a) Map of Canada highlighting the study site marked with a red dot, and (b) areal**
**contours of the two lakes under study. Aerial photography by Madeleine St-Cyr.**

TAS3 has a surface area of 131 m$^2$ and is located on the northern side of the permafrost mound (Table 1, Fig. 1). Surrounded by higher vegetated ramparts, it is shallower and more sheltered from the wind. The average water level fluctuated considerably between the two summer campaigns, reaching 1.80 ± 0.02 m in July 2022 and decreasing to 1.50 ± 0.04 m in August 2023.

TAS3 is more humic than TAS1, with twice the DOC concentration and nearly three times the $a_{320}$ values (Table 2). Its TSS level is also three times higher than in TAS1, making it more opaque, with a $Z_{SD}$ of just 55 cm. Despite these differences, TAS3 exhibited similarly high nutrient concentrations to TAS1 (Table 2).

**Table 1. Morphometric characteristics of the studied thermokarst lakes, derived from bathymetric maps created in July 2022, along**
**with Secchi depth measurements.**

| Lake | Area (m$^2$) | Wind exposure | Max. depth (m) | Mean depth (m) | Secchi depth (m) |
|---|---|---|---|---|---|
| TAS1 | 186 | exposed | 2.5 | 1.2 | 1.23 |
| TAS3 | 131 | sheltered | 1.9 | 0.7 | 0.56 |



**Table 2. Chemical and optical properties of the two lakes surveyed in 2022 and 2023, including the absorption coefficient at 320 nm ($a_{320}$; a proxy for chromophoric dissolved organic matter), dissolved organic carbon (DOC), specific ultraviolet absorbance at 254 nm (SUVA$_{254}$), total nitrogen (TN), total phosphorus (TP), pH, conductivity, and total suspended solids (TSS). Data are presented for both surface and bottom samples. na = not available.**

| Lake | Depth | $a_{320}$ ($m^{-1}$) | DOC (mg $L^{-1}$) | SUVA$_{254}$ (L mg$^{-1}$ m$^{-1}$) | TP ($\mu g\ L^{-1}$) | TN ($\mu g\ L^{-1}$) | pH | Conductivity ($\mu S\ cm^{-1}$) | TSS (mg $L^{-1}$) |
|---|---|---|---|---|---|---|---|---|---|
| | | | | | July 2022 | | | | |
| TAS1 | surface | 8.4 | 4.4 | 2.0 | 48 | 439 | 7.8 | 24.0 | 7.0 |
| | bottom | 3.7 | 2.5 | 1.6 | 350 | 1886 | 6.4 | 40.0 | na |
| TAS3 | surface | 28.1 | 7.4 | 3.8 | 109 | 804 | 6.5 | 20.0 | 22.0 |
| | bottom | 30.4 | 7.1 | 4.0 | 132 | 920 | 6.1 | 21.0 | 26.0 |
| | | | | | August 2023 | | | | |
| TAS1 | surface | 11.6 | 8.6 | 1.5 | 111 | 884 | 6.6 | 24.8 | 16.0 |
| | bottom | 3.1 | 4.9 | 0.8 | 472 | 2139 | 6.1 | 117.0 | 118.0 |
| TAS3 | surface | 38.7 | 10.5 | 3.6 | 159 | 1444 | na | 17.6 | 6.0 |
| | bottom | 40.6 | 7.1 | 5.2 | 446 | 2242 | 5.6 | 46.0 | 40.0 |

## 2.3 Meteorological conditions

To relate oxythermal conditions and GHG fluxes from the lakes to atmospheric conditions, air temperature, wind speed, and wind direction were obtained from a meteorological station located 200 m from the main site (Cen, 2024). In addition, local wind speed was measured 0.2 m above the water surface of both lakes during the intensive summer campaigns using DS-2 Sonic Anemometer (resolution: 0.01 m s$^{-1}$, 1°) mounted on a small floating platform. To estimate the roughness length ($z_0$), we applied the logarithmic wind profile equation using the wind speed measurements taken at 0.2 m above the lakes during the two intensive campaign periods, along with wind speed data from the nearby meteorological station at 10 m height. Calculations were performed under quasi-neutral atmospheric conditions (bulk Richardson number between –0.05 and +0.05), yielding $z_0$ values of 0.087 m at TAS1 and 0.112 m at TAS3.

## 2.4 Limnological profiling

The oxythermal structure was continuously monitored from 7 July 2022 to 21 August 2023 for TAS1 and from 28 September 2021 to 21 August 2023 for TAS3. Dissolved oxygen (PME miniDOT, ±10 µM, ±0.1°C), conductivity (HOBO U24-001, ±1 µS cm$^{-1}$, ±0.1°C), and pressure (HOBO U20L-01, ±0.3% FS, ±0.44°C) were recorded at the surface and bottom only. As these loggers also recorded temperature, water temperature was measured at a total of six depths, including four additional depths with HOBO U22-001 (±0.21°C) (see logger depths in Table S1). The submersible automated data loggers were installed on a mooring line at the deepest point of each lake and recorded at hourly intervals (Fig. 2). The buoy was positioned just below the water level to avoid interference from ice during the winter. As the water level was lower in August 2023, the buoy was slightly above the surface, bringing the loggers closer to the surface than in July 2022.





In addition to continuous data collected by the loggers, manual measurements of temperature, dissolved oxygen, and conductivity were conducted. In 2022, data were recorded using a portable YSI ProODO device (resolution: 0.1% for dissolved oxygen, 0.1°C for temperature) and a Seven2Go™ conductivity meter (resolution: 0.01 μS cm$^{-1}$). In 2023, measurements were taken with a ProSOLO device (resolution: 0.1% for dissolved oxygen, 0.1°C for temperature, 0.001 mS cm$^{-1}$ for conductivity). The buoyancy frequency ($N$, cycles per hour), which quantifies water column stability, was calculated using the following equation:

$$N = \sqrt{-\frac{g}{\rho}\frac{d\rho}{dz}}\frac{3600}{2\pi} \tag{1}$$

where $g$ is the acceleration due to gravity (9.81 m s$^{-2}$), $\rho$ is the water density (kg m$^{-3}$), and $z$ is the depth (m). To quantify the stabilizing effect of stratification relative to the destabilizing effect of wind-induced shear, the Wedderburn number ($W$) was calculated as:

$$W = \frac{g\,\Delta\rho\,h^2}{\rho\,u_{*w}^2\,L} \tag{2}$$

where $\Delta\rho$ is the density difference across the diurnal thermocline (kg m$^{-3}$), $h$ is the depth of the thermocline (m), $\rho$ is the average water density (kg m$^{-3}$), $u_{*w}$ is the friction velocity (m s$^{-1}$), and $L$ is the length of the lake in the wind direction (m). Values of $W > 10$ indicate minimal thermocline tilting, $1 < W < 10$ indicate partial tilting, and $W < 1$ indicate full upwelling.

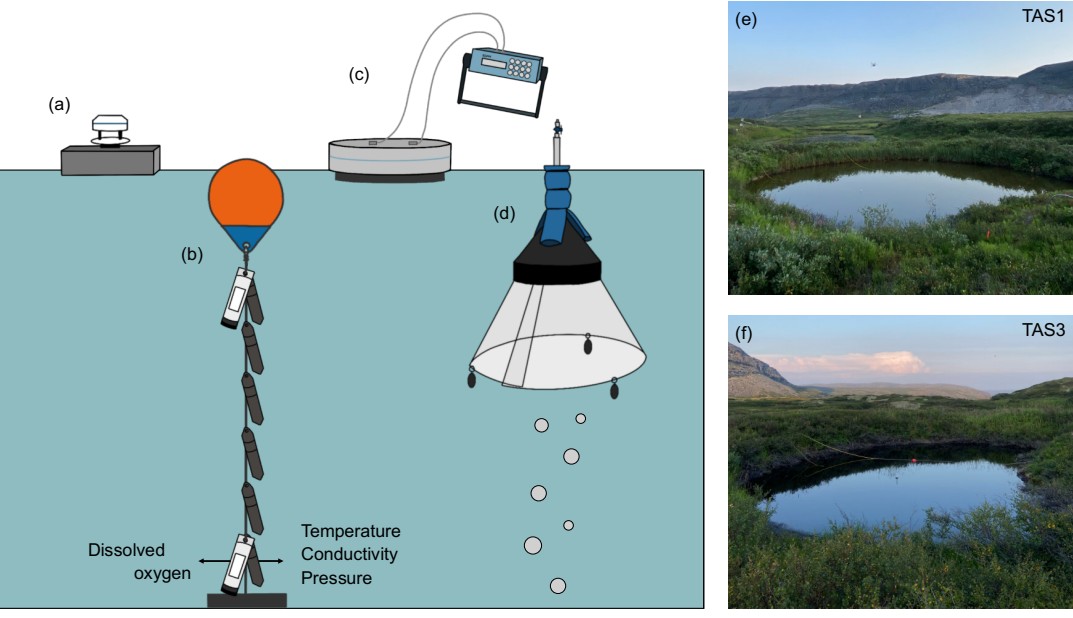

**Figure 2. Schematic of the instrumentation deployed in the lakes, including: (a) an anemometer mounted on a floating platform (deployed in July 2022 and August 2023), (b) a mooring line equipped with data loggers for continuous oxythermal measurements, (c) a floating chamber connected to an infrared CO$_2$ analyzer (deployed in July 2022 and August 2023), and (d) a bubble trap (deployed in August 2023). Panels (e) and (f) show photographs of TAS1 and TAS3, respectively.**



## 2.5 Water chemistry

Several limnological characteristics were measured in each lake during the intensive campaigns, including TP, TN, TSS, pH, conductivity, DOC, and CDOM. TP and TN were quantified from unfiltered water samples collected in 50 mL plastic tubes, which were acidified with $H_2SO_4$ to a pH of 2 and stored at 4°C until analysis by colorimetry following digestion (details in Bartosiewicz et al. (2016)). For CDOM and DOC, water samples were filtered through pre-rinsed polyethersulfone (PES) filters (0.2 µm porosity) and stored in amber glass vials at 4°C. For DOC analysis, 40 µL of pure HCl was added to the vials to adjust the pH to 2 prior to analysis using a Total Organic Carbon analyzer (Aurora 1030W, O.I. Analytical) with the persulfate oxidation method. CDOM samples were brought to room temperature within a week of field collection, and absorbance scans were conducted between 200 and 800 nm using a dual beam spectrophotometer (Varian Cary 100 Bio UV-Visible, ± 2 nm) with a 1-cm pathlength quartz cuvette. Null-point adjustment was applied to blank-corrected absorbance spectra using the mean value from 790 to 800 nm. The absorption coefficient at 320 nm ($a_{320}$) was used as a proxy for CDOM concentration. For TSS, water samples were filtered onto pre-burned and pre-weighed glass fiber filters (GF75, 0.7 µm nominal porosity, Advantec). The filters were stored in aluminum foil in the freezer and then dried overnight at 60°C to determine TSS content.

## 2.6 GHG flux measurements and estimations

To estimate GHG fluxes, we measured the dissolved concentrations of $CO_2$, $CH_4$, and $N_2O$, as well as $CO_2$ diffusive fluxes using a floating chamber. Ebullitive fluxes of $CO_2$ and $CH_4$ were measured using an inverted funnel submerged below the water surface. $CO_2$-equivalent emissions were calculated by applying a global warming potential (GWP) values of 27 for $CH_4$ and 273 for $N_2O$, based on a 100-year time horizon, with $CO_2$ having a GWP value of 1 (IPCC, 2023). All data are presented in Eastern Standard Time (EST).

### 2.6.1 Dissolved gas concentrations

Dissolved gas concentrations were obtained at the lake surface during each diffusive flux measurement using the headspace method. To account for the accumulation of GHG in the water column, gases were also collected at various depths. In 2022, four profiles were carried out (10, 80, 160, and ~230 cm at TAS1; 10, 50, ~105, and ~155 cm at TAS3). In 2023, two profiles were conducted (10, 120, and 230 cm at TAS1; 10, 70, and 140 cm at TAS3). Water samples were collected from the centre of the lakes using a Van Dorn horizontal sampler (integrating a depth layer of ~15 cm). The water was immediately transferred to a 2 L LDPE gas exchange bottle, except at the surface, where it was sampled directly into the bottle. A 30 mL headspace of atmospheric air was created using a syringe connected to an inlet at the bottom of the bottle. The bottle was shaken vigorously for 3 min, and 20 mL of the gaseous headspace was transferred to a 12 mL gas-tight Exetainer vial that had been pre-flushed with nitrogen and vacuumed. Water temperature was measured at the beginning and end of the exchange process. Atmospheric samples were also taken daily. Gas samples were analysed using a Trace 1310 gas chromatograph (Thermo Fisher Scientific;



column capacity 4, TCD detector). Calibration curves were established for $CO_2$ (up to 10 000 ppm), $CH_4$ (up to 45 000 ppm), and $N_2O$ (up to 1 ppm). Detection limits were 200 ppm for $CO_2$, 3 ppm for low $CH_4$ concentrations, 50 ppm for $CH_4$ concentrations above 1000 ppm, and 0.1 ppm for $N_2O$. Dissolved $CH_4$ and $CO_2$ concentrations were calculated using Henry's
Law, adjusted for temperature and atmospheric pressure.

### 2.6.2 Diffusive fluxes

Diffusive fluxes ($F_D$) of $CO_2$ (mmol m$^{-2}$ d$^{-1}$) were measured over a full diel cycle during both intensive campaigns (18 measurements in 2022 and 24 in 2023; see Table S2). These fluxes were obtained by monitoring gas concentration in a floating chamber (area: 0.147 m$^2$, volume: 0.0128 m$^3$) connected to an infrared gas analyser (EGM4, PP Systems) for approximately
30 min, following the method of Macintyre et al. (2021):

$$F_D = \frac{S\,P\,V_{ch}}{R\,T\,A_{ch}}\,D \qquad (3)$$

where $S$ is the linear slope of $CO_2$ (ppm) versus time, $P$ is the atmospheric pressure (Pa), $V_{ch}$ is the chamber volume (m$^3$), $R$ is the gas constant (J mol$^{-1}$ K$^{-1}$), $T$ is the water temperature (K), $A_{ch}$ is the chamber surface area (m$^2$), and $D$ is a unit conversion factor to obtain flux values in mmol m$^{-2}$ d$^{-1}$.

To estimate the $CH_4$ and $N_2O$ fluxes, we used the concentration gradient and the gas transfer velocity ($k$):

$$F_D = k\left(C_{sf} - C_{eq}\right) \qquad (4)$$

where $C_{sf}$ is the gas concentration at the water surface (obtained from the headspace method), and $C_{eq}$ the aqueous gas concentration in equilibrium with the atmosphere at ambient temperature, both in µM. From this equation, we obtained the $k$ value for $CO_2$ using direct flux measurements from the chamber and surface water $CO_2$ concentrations. The corresponding $k$
values for $CH_4$ and $N_2O$ were computed using the following equation:

$$k_x = k_{CO_2}\left(\frac{Sc_x}{Sc_{CO_2}}\right)^{-n} \qquad (5)$$

where $Sc_x$ is the Schmidt number (the ratio of kinematic viscosity and molecular diffusion) for gas $x$ at ambient temperature. The exponent $n$ was set to 1/2 when wind speeds exceeded 3.6 m s$^{-1}$, and 2/3 when wind speeds were below 3.6 m s$^{-1}$ (Jähne et al., 1987).

### 240   2.6.3 Ebullitive fluxes

Ebullitive fluxes ($F_E$) of $CO_2$ and $CH_4$ were measured in 2023 following the method outlined by Wik et al. (2013). Bubble traps, consisting of inverted funnels with a collection area of 0.23 m$^2$, were installed for a total 11 days in both lakes (Fig. 2). $N_2O$ concentrations could not be quantified due to interference from high $CH_4$ concentrations. The bubble traps were connected to a 140 mL graduated polypropylene syringe and were sampled every 1 to 4 days, depending on the gas accumulation rate.



Gas was collected in two replicates (Exetainers) for quantification by gas chromatography. Ebullitive fluxes ($F_E$) were calculated using the following equation:

$$F_E = \frac{P\, V_g}{\Delta t\, V_m\, A} \tag{6}$$

where $P$ is the partial pressure of the gas (dimensionless), $V_g$ is the total volume of gas in the funnel (m$^3$), $\Delta t$ is the time interval between two measurements (days), $V_m$ is the molar volume of gas at local air temperature (m$^3$ mol$^{-1}$), and $A$ is the collection

area of the funnel (m$^2$).

**2.7 Larger scale estimates**

To estimate annual GHG diffusive emissions from the lakes, we calculated the average, minimum, and maximum fluxes for summer 2022, fall 2022, spring 2023, and summer 2023. For each season, Equation 4 was applied using the minimum and maximum concentration gradients estimated for that season, along with the average gas transfer velocity ($k$) over the

corresponding period. We estimated $k_{600}$, the gas transfer velocity normalized to a Schmidt number of 600, to calculate $k$ with equation 5. Limnological seasons were defined based on changes in the thermal structure of the water column. For summer comparisons, the period from 7 July to 20 August was used, as data were available for both 2022 and 2023. Fall 2022 and spring 2023 were defined as the periods from September to October and mid-May to mid-June, respectively.

To estimate the concentration gradients for each season, different approaches were applied based on data availability. For the

two summers, we used the average surface concentration measured during the two-week intensive campaigns. For fall 2022, in the absence of direct measurements, we estimated the summer storage flux from the dissolved gas concentration profiles recorded during the intensive campaigns. Given the lakes are very small, it was assumed that the GHG accumulating in deeper layers would have sufficient time to mix with surface waters and equilibrate with the atmosphere before ice cover formation. Thus, the theoretical surface concentration was calculated by mixing the water column gas concentrations, using the volume

of each water layer derived from hypsographic curves, as described in Prėskienis et al. (2021). For spring 2023, we estimated surface concentrations using the ratio between summer and winter gas concentrations observed by Matveev et al. (2019), as no measurements under the ice were available to calculate the winter storage flux. Matveev et al. (2019) studied $CH_4$ and $CO_2$ concentrations beneath the ice of five thermokarst lakes in late winter and compared them with summer concentrations. These lakes are similar to those in this study, located in the subarctic region of Quebec, in peatland valleys near the

Sasapimakwananisikw River. The lakes are shallow (maximum depth of 1.4 to 2.8 m) and contain high concentrations of organic matter, resulting in limited water transparency. The median ratios of dissolved gas accumulated in the water column (in mmol m$^{-2}$) for winter relative to summer were 0.67 for $CO_2$ and 1.06 for $CH_4$. These ratios were used to estimate storage concentrations, assuming that all GHGs accumulated under the ice would mix with surface waters and equilibrate with the atmosphere during spring before summer stratification. This assumption may not fully reflect reality, as Matveev et al. (2019)

observed a very short mixing period in spring, with stratification quickly establishing after the ice cover disappeared, leaving





only the upper part of the water column to release its gas content. Nevertheless, we considered this as a *potential* spring storage flux.

The average seasonal gas transfer coefficient $k_{600}$ (cm h$^{-1}$), was calculated using the small eddy version of the surface renewal model (SRM) (Macintyre et al., 1995), which accounts for both heat exchange processes and wind-induced turbulence at the
air-water interface:

$$k_{600} = c_1 (\varepsilon v)^{\frac{1}{4}} S_c^{-\frac{1}{2}} \tag{7}$$

where $v$ is the kinematic viscosity (m$^2$ s$^{-1}$), $c_1$ is an empirically derived coefficient, $\varepsilon$ is the dissipation rate of turbulent kinetic energy (m$^2$ s$^{-3}$), and $S_c$ is the Schmidt number defined above. The dissipation rates driven by wind shear ($u_{*w}$) and buoyancy flux under cooling ($\beta$) were computed following Tedford et al. (2014). Surface energy fluxes were derived from meteorological
data and surface water temperatures recorded by automated thermistors installed in the lakes. The depth of the surface layer (i.e., the mixing epilimnion) was determined based on the rapid decline in dissolved oxygen (Fig. 5), which varied between years. In the summers 2022 and 2023, the surface layer depths were estimated at 2.1 m and 1.0 m for TAS1, and 1.1 m and 0.5 m for TAS3, respectively. During fall 2022 and spring 2023, the surface layer was assumed to extend to the bottom of the lakes. As noted in previous studies, uncertainty in the parameter $c_1$ limits the application of the SRM for estimating gas fluxes
(Read et al., 2012; Zappa et al., 2007; Macintyre et al., 2010; Vachon et al., 2020; Macintyre et al., 2021). Theoretical and experimental studies suggest that $c_1$ typically approximates 0.4 (Zappa et al., 2007; Lamont and Scott, 1970), but can range between 0.2 and 1.0 depending on turbulence intensity, water depth, and surface cleanliness (Macintyre et al., 2018; Wang et al., 2015). Given that our lakes are smaller and shallower than those in previous studies, we adopted the lower boundary value of $c_1 = 0.2$.

## 295 3 Results

We first present the annual variations in temperature, oxygen, and conductivity observed in the two thermokarst lakes. We then detail the results from the two intensive campaigns, including meteorological conditions, physicochemical profiles, and GHG concentrations and fluxes, to examine the magnitude and diel variations during the summer. Finally, we provide broader estimates comparing GHG fluxes across summer, fall, and spring.

## 300 3.1 Seasonal profiles

Both lakes have an open-water period of approximately five months (June to October) and remain ice-covered for about seven months (November to May) (Fig. 3). Ice cover duration was determined using Sentinel-2 imagery analysis with an ice detection algorithm through the Google Earth Engine platform, following the approach of Domart et al. (2024). In the winter preceding the 2022 summer campaign, ice began forming on both lakes on 10 November 2021, with ice breakup occurring on 6 June
2022 for TAS1 and 3 June 2022 for TAS3. In the following winter, ice cover lasted from 10 November 2022 to 6 June 2023





for TAS1, whereas TAS3 experienced earlier ice breakup on 19 May 2023. During summer, both lakes undergo partial mixing events due to their sensitivity to temperature fluctuations, causing stratification to weaken rapidly and making them more susceptible to wind-driven mixing. However, complete mixing only occurs in autumn and spring, classifying them dimictic lakes, although the spring mixing period is notably brief.



**Figure 3. Physicochemical profiles from summer 2022 to summer 2023, showing temperature and buoyancy frequency *N* for (a) TAS1 and (c) TAS3, with the water surface marked by a black dotted line. Oxygen saturation and specific conductivity (SC) are shown for (b) TAS1 and (d) TAS3. Temperature at the very surface was not measured, as the buoy was positioned below the air-water interface to prevent it from becoming trapped in the ice cover. Nevertheless, winter temperature data suggest that ice may have grown over the buoys, pushing down the mooring and changing the depth of surface sensors, making surface data unreliable. Intensive summer campaign periods are indicated by brackets for July 2022 and August 2023.**



At TAS1, the mooring was installed from 7 July 2022 to 21 August 2023, so data from the spring mixing period and the onset of stratification before the 2022 summer campaign are unavailable. However, air temperature data indicate a colder period between late June and early July 2022, with an average daily air temperature of 6.2°C from 23 June and 11 July (Fig. S2). This

likely led to full oxygenation of the bottom waters by mid-July, as shown by the bottom oxygen logger positioned at 58 cm above the lake bottom (logger depths in Table S1). After a brief period of stratification at the end of July 2022, marked by a peak $N$ value of 64 cycles per hour (cph; black curve in Fig. 3a), a colder spell from 4 to 11 August 2022 weakened stratification, causing $N$ to drop to a minimum of 7.5 cph and resulting in partial mixing of the water column. Stratification was then re-established, reaching a maximum $N$ of 57.1 cph on 23 August, before another partial mixing event led to a decline

in $N$ to 6.5 cph until 8 September. Autumnal mixing began on 18 September 2022. Although ice cover analysis indicates ice formation began on 10 November 2022, reverse stratification was observed from 2 November. The bottom waters became fully anoxic by 12 November, while surface anoxia was not detected until 9 December. Ion accumulation under the ice throughout the winter increased conductivity to a maximum of 124 μS cm$^{-1}$ before it dropped on 12 May 2023, likely due to partial ice melt. Full ice-off occurred by 6 June 2023. Temperature, conductivity, and oxygen profiles suggest that the spring

mixing period in TAS1 was extremely brief, lasting only about five days (1-5 June 2023, Fig. 3a,b), but oxygen levels remained near anoxia. Surface oxygen began to increase on 5 June, yet bottom anoxia returned as early as 8 June. Thermal stratification persisted throughout the summer, more pronounced than in 2022, maintaining bottom anoxia. During the comparable summer window (7 July - 21 August), the average $N$ value was 44.6 cph in 2023, compared to 34.4 cph in 2022. Oxygen data from 26 June to 14 July 2023 were deemed unreliable for TAS1 and had to be discarded. However, the thermal structure suggests that

anoxic conditions likely persisted during this period (indicated by the red dotted line in Fig. 3b).

At TAS3, the mooring was deployed for nearly two years, from 28 September 2021 to 21 August 2023. During the first recorded winter in 2022, ion accumulation under the ice increased conductivity to a maximum of 123 μS cm$^{-1}$ before it dropped on 11 May 2022, likely due to partial ice melt. The spring mixing period lasted approximately 23 days (11 May - 3 June; Fig. 3c,d), as indicated by conductivity and oxygen data. However, full ice-off did not occur until 3 June, and temperature profiles

show that the water column only began warming from 1 June, keeping surface waters low in oxygen. Thermal stratification established quickly on 3 June 2022, leading to rapid bottom-water deoxygenation, with complete anoxia occurring within five days (Fig. 3c,d), as recorded by the oxygen logger positioned at 44 cm above the sediment layer (Table S1). Between 25 June and 10 July 2022, weaker stratification was observed, with $N$ values reaching a minimum of 7.5 cph (black curve in Fig. 3c). Partial mixing events on 25 June and 4 July partially reoxygenated the bottom waters. Stable stratification re-established

thereafter, reaching a maximum $N$ of 80.4 cph and persisting until a partial mixing event on 5 August. Strong stratification continued through the end of August, with intermittent weakening thereafter, but bottom waters remained anoxic until 23 September 2022. Autumnal turnover began in late September, yet bottom water reoxygenation remained incomplete before ice formation. As observed at TAS1, inverse stratification set in on 2 November, leading to full bottom-water anoxia by 7 November, which progressively reached the surface by 27 November 2022. Throughout the winter, ion accumulation under

the ice raised conductivity to a maximum of 110 μS cm$^{-1}$ before it dropped on 19 May 2023 following ice breakup. At this



time, temperature, conductivity, and bottom-oxygen data indicate that the spring turnover lasted approximately 13 days in TAS3 (19 May - 1 June 2023, Fig. 3c,d). However, stratification developed more rapidly than in spring 2022, leading to near-instantaneous bottom anoxia by 9 June 2023. As for TAS1, thermal stratification persisted throughout summer in 2023 and was stronger than in 2022. During the comparable summer window (7 July - 21 August), the average $N$ value was 50.8 cph in 2023, compared to 41.2 cph in 2022. From 8 July 2023 onward, bottom-water conductivity progressively increased, likely due to the stronger and more persistent stratification in this warmer year.

### 3.2 Meteorological conditions

Overall, summer 2022 was colder than summer 2023, with an average air temperature of 11.3 ± 5.0°C for June, July and August 2022 (0.7°C above seasonal normal) compared to 13.1 ± 4.4°C in 2023 (2.6°C above seasonal normal). Since the intensive campaigns took place at different periods in each summer, meteorological comparisons are made between July 2022 and August 2023.

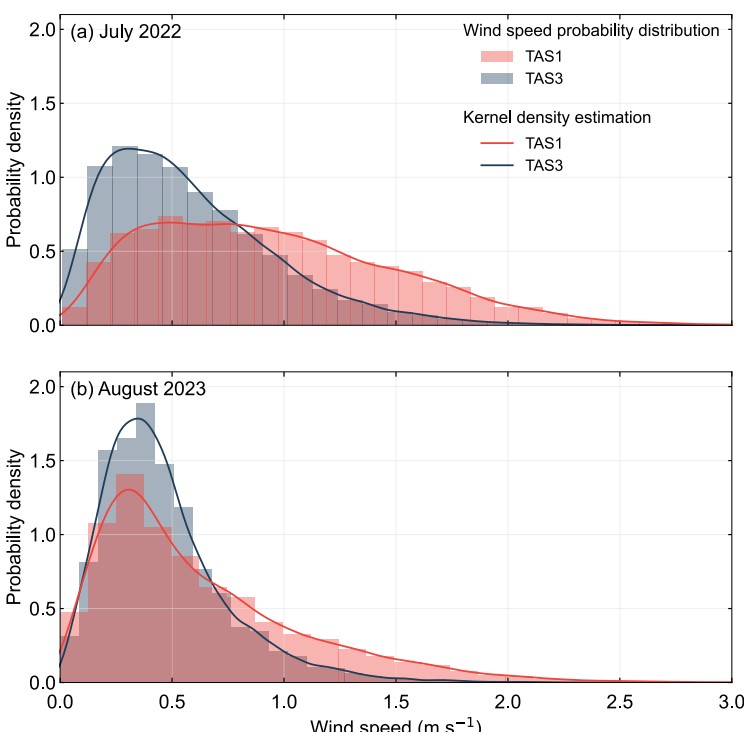

**Figure 4. Wind speed probability density function and kernel density estimation at the surface (0.2 m from surface) of TAS1 and TAS3 during the intensive summer campaigns of (a) 7-16 July 2022 and (b) 9-21 August 2023.**



Weather conditions varied considerably between these two sampling periods, with July 2022 being significantly colder and windier than August 2023. The mean air temperature during the July 2022 campaign was 8.8°C (range: 1.3-23.1°C), whereas in August 2023, it was substantially warmer at 14.6°C (range: 6.6-23.4°C). The mean wind speed at 10 m height was $4.0 \pm 1.7$ m s$^{-1}$ in 2022, compared to $3.5 \pm 2.0$ m s$^{-1}$ in 2023. Closer to the water surface (0.2 m above), wind speeds were on average 1.4 times higher in 2022 than in 2023. Wind conditions also varied between the sampling sites, with TAS1 experiencing wind speeds 1.5 times higher than TAS3 on average (Fig. 4), as TAS3 is more sheltered by surrounding vegetation and topography. In both years, prevailing surface winds of TAS1 were mainly from the north and west, whereas winds at TAS3 predominantly came from the west (Fig. S3).

### 3.3 Vertical structure during the intensive sampling periods

Both lakes exhibited thermal stratification during the two intensive observation periods (indicated by brackets in Fig. 3). Figure 5 presents individual vertical profiles collected manually from the lakes. Temperature profiles (Fig. 5a,d) indicate that stratification was most pronounced in August 2023, due to warmer conditions and the later timing within the open water season (by one month). Despite their shallow depths (2.5-2.6 m at TAS1 and 1.5-1.8 m at TAS3), both lakes exhibited oxygen stratification, with anoxic conditions at the bottom during both observation periods (Fig. 5b,e).

Prior to the 2022 intensive campaign, the lakes had recently undergone a 15-day partial mixing event (25 June - 10 July), as indicated by the weaker stratification and bottom-layer reoxygenation at TAS3 (Fig. 3c,d). This mixing event likely released accumulated gases from the previous winter and stratification period. During this time, mixed layers were significantly deeper than in August 2023. Based on the rapid decline in dissolved oxygen marking the boundary of the isolated layer, mixed layers extended to 2.1 m at TAS1 and 1.1 m at TAS3 (Fig. 5). These deeper mixed layers allowed a larger portion of the water column to interact with the atmosphere, resulting in greater oxygenation. However, below the mixed layer, oxygen remained depleted, suggesting the accumulation of GHGs.

In contrast, during the August 2023 campaign, strong thermal stratification prevailed, leading to anoxic bottom waters from early June in both lakes. The mixed layers were much shallower–only 1.0 m at TAS1 and 0.5 m at TAS3. This stronger thermal stratification was reflected in the sharper oxygen concentration gradient at TAS3. Greater light attenuation was also observed at TAS3, where $a_{320}$ values were more than three times higher than at TAS1 in both years (Table 2). Additionally, specific conductivity at the bottom showed considerable differences in 2023. While surface remained similar, conductivity increased with depth, especially at TAS1 (up to 128 µS cm$^{-1}$), contributing to stronger density stratification.

Despite the small fetch of these lakes (~15 m at TAS1 and ~12 m at TAS3), our estimations revealed low Wedderburn numbers ($W$) during the day in both intensive campaigns (Fig. S4), indicating susceptibility to mixing as wind increased. During the 2022 intensive campaign, $W$ mostly indicated partial tilting ($1 < W < 10$) during the day at TAS3, whereas at TAS1, they generally suggested minimal tilting ($> 10$). By mid-July (from 12 July), values suggested minimal tilting in both lakes, particularly at night. In August 2023, $W$ mostly indicated partial tilting during the day but minimal tilting at night (reaching up to $10^8$ on 15 August). Episodes of full upwelling ($W < 1$) were estimated during the day on 11 and 17 August TAS3. Throughout



both intensive campaigns, TAS3 consistently showed lower $W$ values, suggesting a greater potential for tilting compared to TAS1.

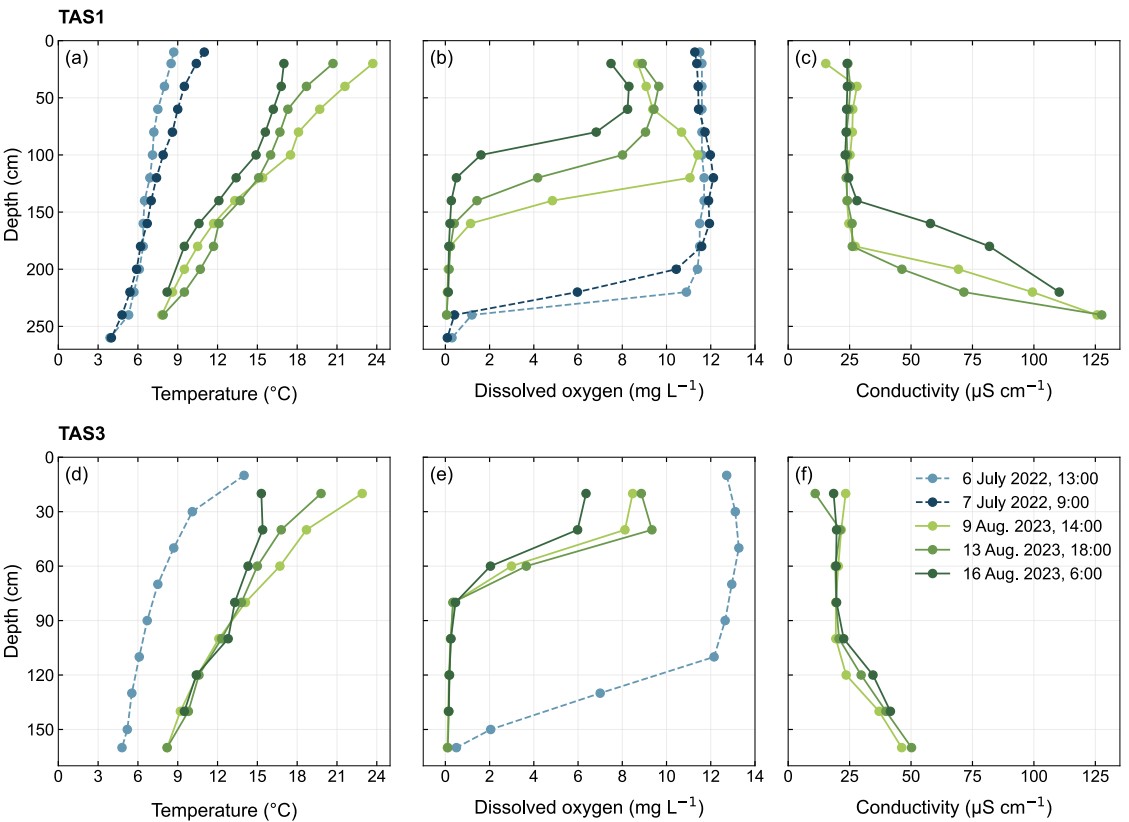

**Figure 5. Vertical profiles of temperature, dissolved oxygen and specific conductivity in TAS1 (top panels) and TAS3 (bottom panels). Dotted lines are profiles in 2022, and solid lines are profiles in 2023.**

### 3.4 Summer GHG concentrations

During both intensive summer campaigns, the lakes were supersaturated with $CO_2$ and $CH_4$ at the bottom but mostly undersaturated with $N_2O$ (Fig. 6). In the July 2022 campaign, increased accumulation of GHGs in the bottom layers resulted in a pronounced vertical gradient of gas concentrations. Specifically, $CO_2$ concentrations at the bottom were, on average, 11 times higher than at the surface, while the $CH_4$ gradient was even more pronounced, with bottom concentrations 144 times greater than at the surface on average. In contrast, $N_2O$ concentrations decreased with depth, with surface values on average 1.2 times lower than those in the deeper layers. These gradients resulted in surface gas concentrations that were mostly above air equilibrium for $CO_2$, $CH_4$ and $N_2O$ (Table 3). To assess gas saturation, we used global atmospheric partial pressure values, which averaged 20.1 μM for $CO_2$, 0.0034 μM for $CH_4$, and 12.0 nM for $N_2O$ (grey section, Fig. 6). Moreover, differences in



surface dissolved gas concentrations were observed between the two lakes. $CO_2$ and $CH_4$ surface concentrations were respectively 1.5 and 2.9 times higher at TAS1 compared to TAS3, despite similar bottom concentrations of both gases and a lower pH at TAS3, which could have favored $CO_2$ solubility (Table S3, Table 2).

**Table 3. Near-surface $CO_2$, $CH_4$ and $N_2O$ concentrations measured at TAS1 and TAS3. Values are presented as mean ± standard deviation (min-max). $N$ =10 for 2022 and $N$ =13 for 2023.**

| Period | Surface $CO_2$ (µM) | | Surface $CH_4$ (µM) | | Surface $N_2O$ (nM) | |
|---|---|---|---|---|---|---|
| | TAS1 | TAS3 | TAS1 | TAS3 | TAS1 | TAS3 |
| July 2022 | 37.0 ± 5.5 (27.0 - 43.3) | 25.0 ± 7.5 (10.2 - 34.3) | 1.3 ± 0.4 (0.9 - 2.3) | 0.4 ± 0.1 (0.3 - 0.6) | 13.9 ± 0.9 (12.5 - 15.2) | 13.5 ± 1.1 (11.9 - 14.9) |
| August 2023 | 51.1 ± 32.6 (21.4 - 122.5) | 94.2 ± 75.0 (28.5 - 236.2) | 1.8 ± 0.7 (1.1 - 3.2) | 3.5 ± 2.4 (1.1 - 9.9) | 4.0 ± 0.4 (3.3 - 4.6) | 3.3 ± 0.3 (2.7 - 3.7) |

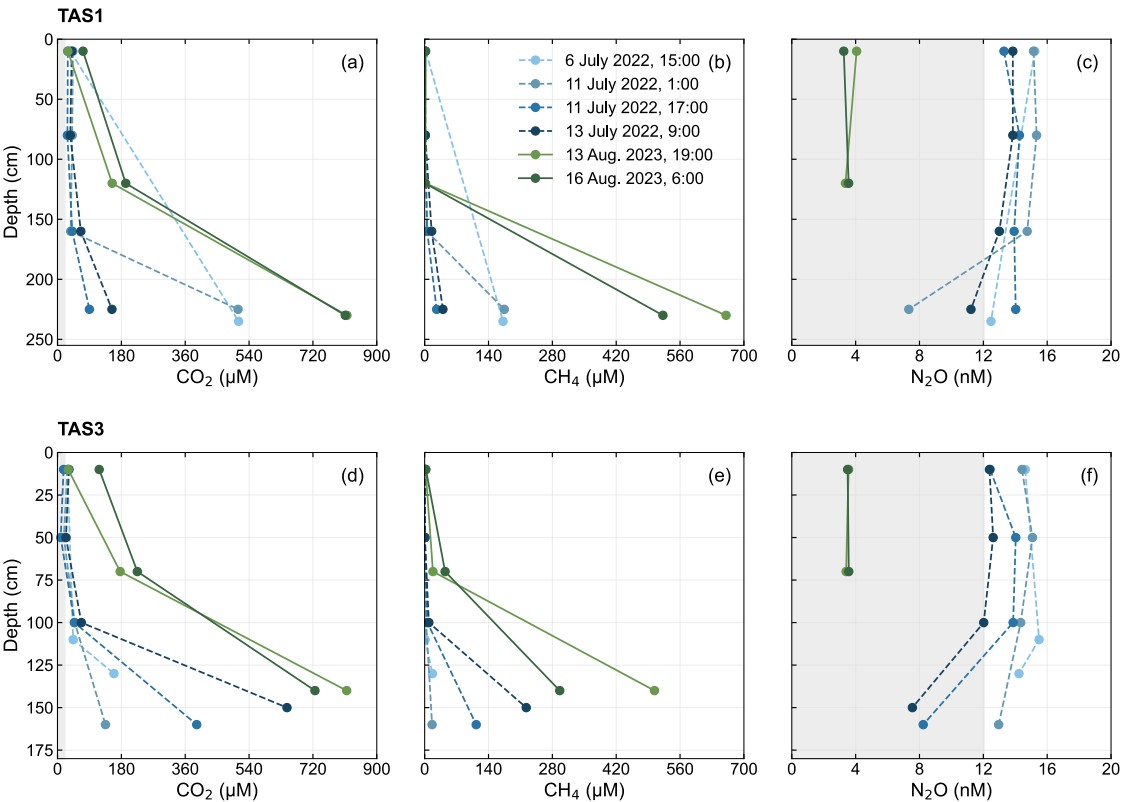

**Figure 6. Concentration profiles of $CO_2$, $CH_4$ and $N_2O$ from TAS1 (top panels) and TAS3 (bottom panels). Dotted lines represent**
**profiles from 2022, while solid lines represent profiles from 2023. The shaded area represents the concentrations below atmospheric partial pressure values.**



During the August 2023 campaign, GHG accumulation in the bottom layers led to a steeper vertical gradient. $CO_2$ concentrations at the bottom were, on average, 12 times higher than at the surface, while bottom $CH_4$ concentrations were 223 times higher than at the surface. $N_2O$ bottom concentrations were not available in 2023. These gradients resulted in surface gas concentrations that were strongly above air equilibrium for $CO_2$ and $CH_4$ but below air equilibrium for $N_2O$ (Table 3). When comparing the two lakes, $CO_2$ and $CH_4$ surface concentrations at TAS3 were approximately twice those found at TAS1, despite lower bottom concentrations at TAS3. This difference was again accompanied by a lower pH at TAS3 (Table S3, Table 2).

Overall, the lakes accumulated 2.4 times more $CO_2$ and 5 times more $CH_4$ at the bottom in August 2023 compared to July 2022. While bottom $N_2O$ concentrations are unavailable for 2023, concentrations in the middle of the water column were already 3 times lower than those observed in 2022. Surface concentrations of $CO_2$ and $CH_4$ were also higher in August 2023 compared to July 2022, while $N_2O$ levels were much lower.

## 3.5 Summer GHG fluxes

In both years, $CO_2$ and $CH_4$ exhibited variable fluxes, with emissions predominantly positive, while $N_2O$ fluxes remained consistently close to equilibrium.

In 2022, $CO_2$ diffusive fluxes ranged from –2.0 to 17.1 mmol m$^{-2}$ d$^{-1}$, while $CH_4$ fluxes ranged from 0.1 to 1.1 mmol m$^{-2}$ d$^{-1}$, and $N_2O$ fluxes oscillated around zero, never exceeding 0.0036 mmol m$^{-2}$ d$^{-1}$ (Table 4). In terms of $CO_2$-equivalent emissions, $CO_2$ accounted for the largest portion of average diffusive emissions, contributing nearly 56% of the total, followed by $CH_4$ at 41%, and $N_2O$ at only 3.0% (Fig. 7). This analysis excludes ebullition, as it was not assessed in 2022. Average diffusive GHG emissions from TAS1 were five times higher than those from TAS3, consistent with the higher surface concentrations observed at TAS1.

In 2023, $CO_2$ diffusive fluxes ranged from 7.8 to 67.4 mmol m$^{-2}$ d$^{-1}$, $CH_4$ fluxes ranged from 0.3 to 7.2 mmol m$^{-2}$ d$^{-1}$, and $N_2O$ fluxes ranged from –13.2 to –0.7 mmol m$^{-2}$ d$^{-1}$ (Table 4), resulting from the negative gradients at the air-water interface (Fig. 6). Similar to 2022, $CO_2$ remained the dominant contributor in terms of $CO_2$-equivalent diffusive emissions, comprising 64% of total emissions, while $CH_4$ contributed 39%, and $N_2O$ mitigated total emissions by 4% (sink; Fig. 7). Despite total diffusive fluxes being similar between TAS1 and TAS3 in 2023, surface concentrations of $CO_2$ and $CH_4$ were lower at TAS1 (Table 3). This difference is likely due to more turbulent conditions at TAS1, which enhanced overall emissions. Significant $CH_4$ ebullitive fluxes were recorded at both sites, ranging from 0.5 to 18.1 mmol m$^{-2}$ d$^{-1}$, while $CO_2$ ebullitive fluxes remained negligible (Table 4). At TAS1, $CH_4$ ebullitive fluxes were slightly lower than diffusive fluxes, whereas at TAS3, they were more than six times higher than diffusive fluxes on average. This was also visible at the lake surface, where bubbles accumulated (Fig. S5b). Consequently, ebullition accounted for 70% of total emissions at TAS3, primarily driven by $CH_4$.

Overall, GHG emissions were higher during the intensive campaign period in August 2023 compared to July 2022. The mean diffusive flux of $CO_2$ was 4.7 times higher, while $CH_4$ fluxes were almost four times greater. This increase is attributed to



higher surface concentrations of both gases (Table 3) and greater turbulence, as indicated by the higher $k$ values in 2023 (Table S4). These conditions were reflected on $N_2O$ fluxes, which were consistently negative in 2023, although this flux remains negligible relative to the total flux.

**Table 4. Diffusive fluxes of $CO_2$, $CH_4$, and $N_2O$ in TAS1 and TAS3 ($N$ = 9 in 2022; $N$ = 12 in 2023) and ebullitive fluxes of $CH_4$ in 2023 ($N$ = 4 for TAS1; $N$ = 8 for TAS3). Values are expressed as mean ± standard deviation (min - max). na = not available.**

|  | Lake | July 2022 | August 2023 |
|---|---|---|---|
| $F_D$ $CO_2$ (mmol m$^{-2}$ d$^{-1}$) | TAS1 | 12.1 ± 4.6 (5.7 - 17.1) | 33.6 ± 22.1 (7.8 - 67.4) |
|  | TAS3 | 2.2 ± 2.4 (−2.0 - 4.3) | 33.0 ± 19.6 (9.0 - 66.4) |
| $F_D$ $CH_4$ (mmol m$^{-2}$ d$^{-1}$) | TAS1 | 0.8 ± 0.3 (0.3 - 1.1) | 2.2 ± 1.9 (0.4 - 7.2) |
|  | TAS3 | 0.2 ± 0.1 (0.1 - 0.3) | 1.9 ± 1.1 (0.3 - 3.7) |
| $F_D$ $N_2O$ (μmol m$^{-2}$ d$^{-1}$) | TAS1 | 1.9 ± 1.0 (0.3 - 3.6) | −7.5 ± 3.5 (−13.2 - −2.1) |
|  | TAS3 | 0.8 ± 0.4 (0.4 - 1.4) | −4.5 ± 3.0 (−12.1- −0.7) |
| $F_E$ $CH_4$ (mmol m$^{-2}$ d$^{-1}$) | TAS1 | na | 1.9 ± 1.5 (0.5 - 3.9) |
|  | TAS3 | na | 11.9 ± 4.5 (6.9 - 18.1) |

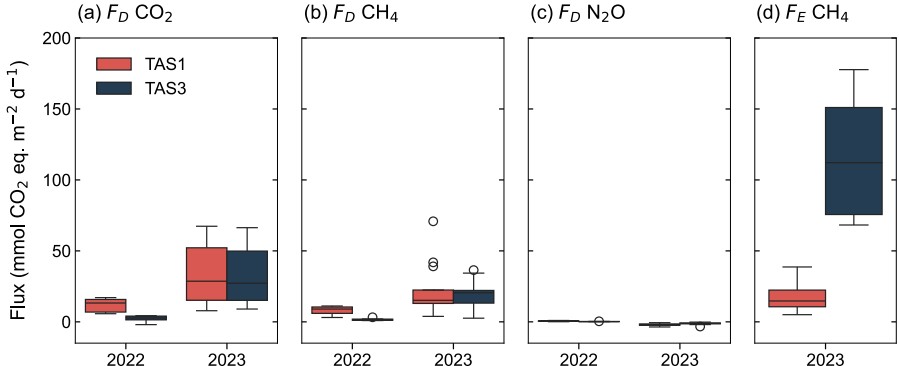

Figure 7. Boxplots of (a) diffusive $CO_2$ flux, (b) diffusive $CH_4$ flux, (c) diffusive $N_2O$ flux, and (d) ebullitive $CH_4$ flux. Fluxes are
470 expressed in mmol $CO_2$ equivalent m$^{-2}$ d$^{-1}$, assuming a global warming potential of 27 for $CH_4$ and 273 for $N_2O$. Ebullitive flux measurements are only available in summer 2023.



## 3.6 Summer diel cycles in stratification and diffusive fluxes

During both intensive summer campaigns, clear diel cycles in the buoyancy frequency ($N$) and dissolved oxygen (DO) were
observed (Fig. 8), as indicated by the data collected from the moorings. Although temperature was more variable in 2022 (Fig. S6), peaks in both $N$ and DO typically occurred in the evening (around 18:00), while the lowest values were recorded in the morning (around 07:00). The grey shaded area in the figure indicates periods of zero solar radiation. All data are presented in Eastern Standard Time (EST).

We compiled surface GHG concentrations from the intensive summer campaigns over several days and calculated fluxes using $k$-estimates from the $CO_2$ chamber method. In 2022, no discernible diel patterns were observed in either the surface concentrations (Fig. S7a,b,c), the gas transfer coefficient (Fig. S8a), or the diffusive emissions of $CO_2$, $CH_4$, and $N_2O$ (Fig. 9a,b,c). However, in 2023, distinct diel cycles for diffusive emissions of all three gases emerged (Fig. 9d,e,f), along with the gas transfer coefficient in both lakes, with higher $k$ values observed during the daytime (Fig. S8b). $CO_2$ concentrations were
elevated in the morning but declined rapidly by noon, likely due to increased photosynthetic activity driven by solar radiation (which peaks around 12:00; Fig. S6) and reduced $CO_2$ solubility as daytime temperatures rose. Similarly, $CO_2$ fluxes exhibited a morning peak between 06:00 and 12:00, followed by a decline to their lowest levels during the evening and night (18:00-03:00; Fig. 9d). $CH_4$ concentrations were generally higher in the morning and lower at night, though this pattern was less pronounced at TAS1, where concentrations were slightly higher during the day. $CH_4$ fluxes also displayed a clear diel pattern,
with daytime peaks between 09:00 and 18:00 and minima recorded at night (21:00-03:00; Fig. 9e). While $N_2O$ concentrations showed no clear diel trend (Fig. S7d,e,f), $N_2O$ fluxes displayed similar variability to $CH_4$, with negative fluxes peaking during the day (09:00-18:00) and approaching equilibrium at night (21:00-03:00; Fig. 9f).



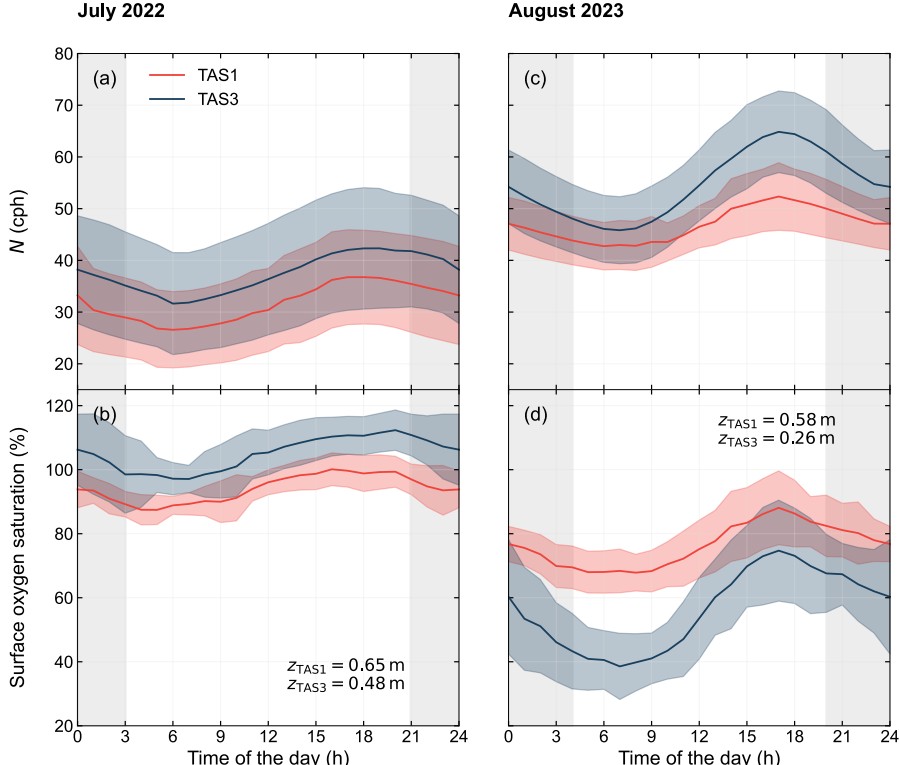

**Figure 8.** Diel cycles of buoyancy frequency $N$ (a,c) and surface dissolved oxygen (in % saturation) at depth $z$ (c,d) for the July 2022 (left) and August 2023 (right) intensive summer campaign periods. The colored shaded area represents the standard deviation, while the grey shaded area indicates periods of zero solar radiation. Time of day is expressed in EST.




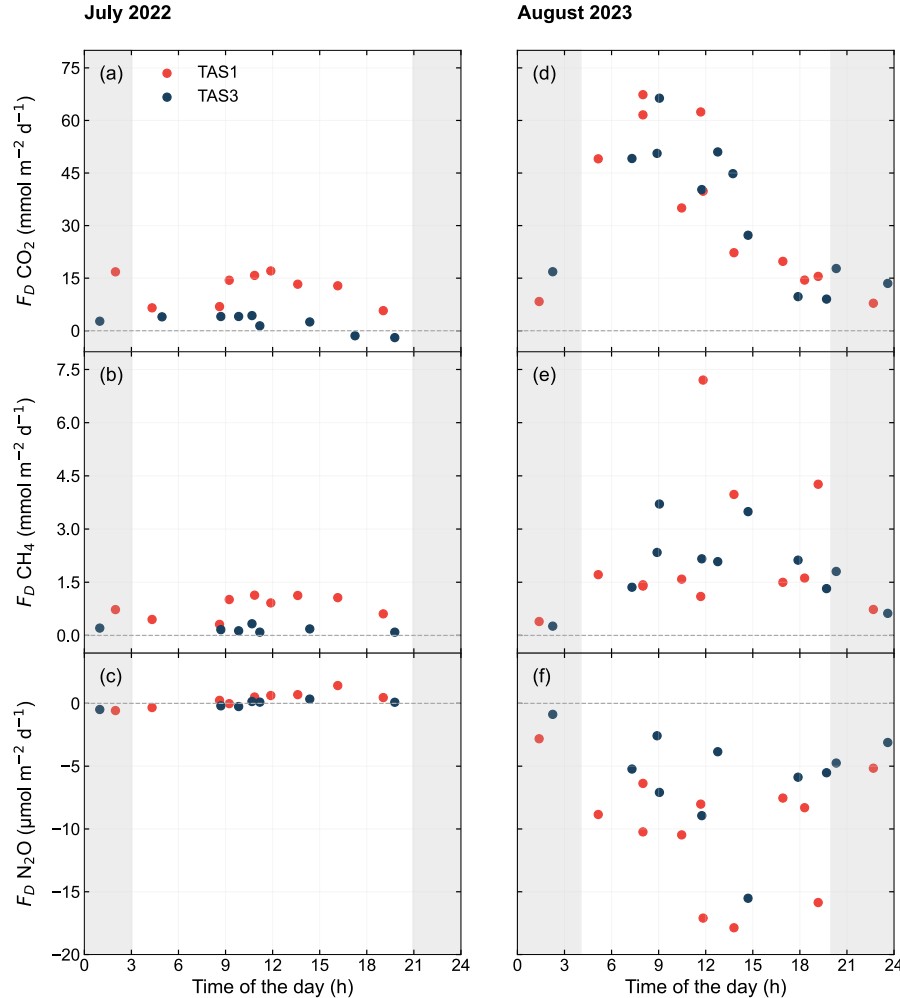

**Figure 9. Diel cycles of $CO_2$ (a,d), $CH_4$ (b,e) and $N_2O$ (c,f) diffusive fluxes for the July 2022 (left) and August 2023 (right) intensive summer campaign periods. $CO_2$ fluxes were measured directly with the floating chamber, while $CH_4$ and $N_2O$ fluxes were obtained by applying the $k$ values derived from $CO_2$ fluxes to their respective concentrations. The grey dotted line represents atmospheric equilibrium. Time of day is expressed in EST.**

## 3.7 Larger-scale estimates

The GHG emissions presented so far are based on direct observations but only cover two short-term intensive sampling periods of two weeks each. To better contextualize these fluxes on a seasonal scale, we extend our analysis to encompass broader temporal variations, acknowledging the inherent uncertainties of this approach. Analysis of seasonal variations in GHG diffusive fluxes from lakes TAS1 and TAS3 reveals significant differences in $CO_2$, $CH_4$, and $N_2O$ emissions across different periods (Table 5). During the summer of 2022, total GHG fluxes were at their lowest. In the fall of 2022, estimations suggest that both lakes could exhibit substantial increases in emissions, with average fluxes more than 4 times higher in TAS1 and 21





times higher in TAS3 compared to those observed in the previous summer. GHG emission estimates continued to rise in the spring of 2023, particularly for $CH_4$, which was 10 times higher in TAS1 and 48 times higher in TAS3 than in summer. During the summer of 2023, flux estimates remained elevated, with average fluxes nearly 1.5 times higher in TAS1 and 10.7 times higher in TAS3 compared to summer 2022, although there was greater variability compared to the previous summer. While

$CH_4$ accounted for an average of 41% and 34% of total GHG emissions estimated in summer 2022 and 2023 respectively, its contribution increased to 74% in autumn and 90% in spring. Overall, the highest fluxes were estimated in spring 2023 at TAS3, with average total fluxes of 243.1 mmol $CO_2$ eq. $m^{-2}$ $d^{-1}$.

**Table 5. Seasonal diffusive flux estimates at lakes TAS1 and TAS3. Values are expressed as min - max for each period. na=not**
**available.**

| Lake | $F_D$ $CO_2$ (mmol $m^{-2}$ $d^{-1}$) | $F_D$ $CH_4$ (mmol $CO_2$ eq. $m^{-2}$ $d^{-1}$) | $F_D$ $N_2O$ (mmol $CO_2$ eq. $m^{-2}$ $d^{-1}$) | GHG (mmol $CO_2$ eq. $m^{-2}$ $d^{-1}$) |
|---|---|---|---|---|
| Summer 2022 | | | | |
| TAS1 | 15.7 (4.7 - 34.4) | 11.5 (3.4 - 24.9) | 0.6 (0.2 - 1.3) | 27.8 (8.2 - 60.6) |
| TAS3 | 5.8 (1.5 - 12.7) | 4.4 (1.1 - 9.7) | 0.5 (0.1 - 1.2) | 10.7 (2.8 - 23.6) |
| Fall 2022 | | | | |
| TAS1 | 31.8 (3.4 - 79.3) | 87.1 (9.3 - 216.5) | –0.3 (–0.0 - –0.8) | 118.6 (12.7 - 295.1) |
| TAS3 | 59.1 (5.8 - 149.5) | 171.5 (16.7 - 432.7) | –0.6 (–0.1 - –1.6) | 230.0 (22.4 - 580.6) |
| Spring 2023 | | | | |
| TAS1 | 9.8 (1.8 - 18.5) | 116.2 (21.4 - 218.3) | na | 126.0 (23.2 - 236.8) |
| TAS3 | 31.0 (5.7 - 62.5) | 212.1 (39.3 - 426.4) | na | 243.1 (45.0 - 488.9) |
| Summer 2023 | | | | |
| TAS1 | 27.7 (4.0 - 63.2) | 14.9 (2.2 - 34.0) | –1.5 (–0.2 - –3.5) | 41.1 (6.0 - 93.7) |
| TAS3 | 80.2 (9.0 - 182.3) | 36.8 (4.2 - 83.6) | –2.2 (–0.3 - –5.0) | 114.7 (12.9 - 260.8) |

## 4 Discussion

The results highlight that GHG emissions from small thermokarst lakes are highly sensitive to interannual variations in meteorological conditions. In this section, we discuss the physicochemical characteristics specific to thermokarst lakes, the summer patterns of dissolved gases and fluxes, and the role of diel and seasonal variability in upscaling lake emissions. Finally,

we reflect on the limitations of the study.

### 4.1 Physicochemistry

The stratified nature of both lakes in summer, despite their modest depth (< 3 m), can be attributed to strong near-surface absorption of solar radiation. The low water clarity, marked by high concentrations of solutes and particles (CDOM and TSS), led to enhanced light absorption at the lake surface. This resulted in pronounced surface warming during the day and evening.





The effect of the summer thermal structure on the oxygen concentration was evident. In 2022, weaker stratification allowed

frequent reoxygenation of the bottom waters during deeper mixing events. This is especially visible at TAS1, though
differences in sensor placement may have influenced observations, with the logger positioned 58 cm above the bottom at TAS1
compared to 44 cm at TAS3. However, oxygen levels quickly returned to anoxic conditions as stability increased (Fig. 3). This
suggests a high oxygen demand from deeper water and lake sediments. Vertical oxygen profiles in July during the intensive

summer campaign showed that much of the water column remain oxygenated, with levels nearing 100% saturation (Fig. 5). In
TAS3, oxygen supersaturation was observed on the afternoon of 6 July, reaching up to 124% near the surface and extending
down to 0.9 m. This reflects active photosynthesis and efficient mixing within the surface layer. In 2023, stronger stratification
isolated the bottom layers of both lakes, making them anoxic throughout the summer. Vertical oxygen profiles in August
during the intensive summer campaign showed that a larger part of the water column remained anoxic. Partial mixing events

eroded this stratification, especially during nighttime cooling, bringing atmospheric oxygen to the surface and oxygen-depleted
water from deeper layers. However, oxygen losses were too substantial, causing surface waters to remain continuously
undersaturated with respect to oxygen that summer (Fig. 3, 5b). A distinct oxygen peak is noticeable in the subsurface layer,
especially in TAS1 on 9 August at 15:00, reaching up to 119% at 100 cm below the water surface. This suggests photosynthesis
and resistance to mixing.

Overall, the differences observed between the two summers underscore the sensitivity of small lakes to interannual variations
in meteorological conditions. Warmer conditions, as observed in 2023, can lead to a rapid onset of spring thermal stratification,
potentially preventing the release of gases accumulated during the winter and maintaining a hypoxic lower water column,
which allows methane to accumulate more rapidly (e.g., Cortés & MacIntyre, 2020). Surprisingly, the two lakes exhibited
significant differences in ice break-up dates. In 2023, ice breakup at TAS1 was estimated on June 6 compared to May 19 at

TAS3. The ice detection algorithm, which relies on Sentinel-2 imagery, may have missed the actual breakup at TAS1 due to
the small size of the lakes and the spatial resolution of 10 m of the imagery. Conductivity data suggest that partial ice melt at
TAS1 occurred in mid-May, though not enough to reoxygenate the water, unlike at TAS3. This discrepancy could be due to
the higher concentration of CDOM in the ice at TAS3, which could have enhanced sunlight absorption and accelerated melting.

**4.2 Summer dissolved gases**

The increased supersaturation of $CO_2$ and $CH_4$ at the bottom of both lakes during the warmer August 2023 sampling campaign,
coupled with the undersaturation of $N_2O$, underscores the key role of thermal stratification in regulating gas concentrations
and vertical gradients. The observed steeper vertical gradients suggest that stratification limited gas evasion to the atmosphere
due to reduced mixing and prolonged isolation of the hypolimnion (Fig. 6). Moreover, the rapid onset of thermal stratification
after ice cover loss in 2023 (Fig. 3) likely trapped some dissolved gases accumulated during the winter. The increased bottom

concentrations of $CO_2$ and $CH_4$ in 2023 may also reflect heightened microbial activity and organic matter decomposition in
the hypolimnion driven by higher temperatures (see below). While bottom $N_2O$ data were unavailable in 2023, the lower mid-





water column $N_2O$ concentrations (Fig. 6) suggest that the expanded anoxic water column created favourable conditions for $N_2O$ consumption (Beaulieu et al., 2014), resulting in negative fluxes.

The greater accumulation of GHGs in the lower layers of both lakes in 2023, along with a shallower mixing layer, increased

the potential for GHG transport to the surface during mixing and penetrating convection. This, in turn, enhanced both the variability and magnitude of surface GHG accumulation compared to 2022. Notably, the higher GHG concentrations at the surface of TAS3 in 2023 may reflect deeper diurnal mixing, as suggested by lower $W$ values during the day ($1 < W < 10$). Since the mixing layer at TAS3 is shallower than at TAS1 (Fig. 5), wind can more easily disrupt stratification. Additionally, the steeper oxygen gradient in this lake may promote $CH_4$ production by extending anoxic conditions over a larger portion of

the water column. Although photosynthesis and respiration influence $CO_2$ concentrations through daytime consumption and nighttime production, the large variability in surface concentrations suggests that these processes alone cannot fully explain the observed differences.

The disparities in surface gas concentrations between the two lakes and intensive campaign periods highlight the complexity of the processes governing gas dynamics in thermokarst lakes. Overall, the enhanced GHG accumulation observed during the

warmer summer suggests that future climate warming could further amplify GHG buildup in stratified lakes, generating larger seasonal variations.

### 4.3 Summer GHG fluxes

Studies presenting GHG fluxes from thermokarst lakes and ponds remain scarce. To compare our results with other studies, Table 6 present the reported $CO_2$ and $CH_4$ flux from different regions. This non-exhaustive list includes only studies that

explicitly identify water bodies formed through thermokarst activity.

In general, our diffusive flux aligns with the range of $CO_2$ and $CH_4$ fluxes reported for lakes and ponds in the region. During the warmer intensive campaign in August 2023, the higher surface concentrations of $CO_2$ and $CH_4$, combined with elevated $k$ values (Fig. S8, Table S4), resulted in significantly higher diffusive fluxes for both gases compared to the colder intensive campaign in July 2022. Thus, the total diffusive flux in August 2023 was 4 times higher than in July 2022. In contrast, $N_2O$

fluxes were notably lower during this period (higher but negative), driven by reduced concentrations and a slightly negative gradient at the air-water interface. Despite the high global warming potential of $N_2O$ (GWP of 273), its uptake had minimal impact on the total GHG fluxes. This pattern is consistent with findings from Davidson et al. (2024), where a shallow eutrophic lake in Denmark acted as a $N_2O$ sink in late summer. Similarly, Bartosiewicz et al. (2016) reported a negligible contribution of $N_2O$ to total GHG emissions from a small, shallow lake in southeastern Canada, where concentrations often remained low

or even undersaturated during a heat wave. In 2022, the higher fluxes at TAS1 corresponded to both higher surface concentrations and higher $k$ values compared to TAS3. However, in 2023, fluxes were similar between the lakes, despite significantly higher surface concentrations at TAS3 (Table 3). This can be attributed to the higher $k$ values at TAS1 (Fig. S8), likely driven by greater exposure to wind, which compensated for its lower surface concentrations. Thus, GHG dynamics can





differ significantly between nearby lakes, and large-scale estimates and comparisons should account for these spatial
differences.

High $CH_4$ ebullition fluxes were recorded in 2023, particularly in TAS3, consistent with findings from other studies
documenting large $CH_4$ ebullition flux during the stratified phase (Davidson et al., 2024). The values observed at TAS3,
particularly humic and sheltered, are among the highest reported for $CH_4$ ebullition in Nunavik (Table 5). However, even
higher $CH_4$ ebullition fluxes have been reported from Arctic thermokarst lakes (Table 6). In terms of $CO_2$-equivalent emissions,
$CH_4$ ebullition at TAS3 contributed 70% of total emissions, compared to 25% at TAS1. The exact reasons for the higher
ebullition rates at TAS3 remain unclear, but several factors may contribute. One possibility is its higher organic content, with
bottom DOC levels nearly three times higher than at TAS1 (Table 2), potentially reflecting a higher microbial productivity.
Plankton was abundant in TAS3 (though not analyzed), and the presence of dense shrub cover along its shores suggests that
falling leaves contribute organic material to the water column and sediment, fueling methanogenic activity (Heslop et al.,
2015). Additionally, it could reflect methodological biases caused by large spatial variations in $CH_4$ bubbling, as highlighted
by Wik et al. (2011) in their study of larger Arctic lakes in Sweden. Although ebullitive fluxes were not measured in 2022, we
hypothesize they were lower, as no visible bubbles were observed at the lake surface, in contrast to the frequent bubbling seen
in 2023 on calm days (Fig. S5). Moreover, the bottom water temperature at TAS1 was 2.1°C colder in July 2022 compared to
August 2023 (5.7°C vs 7.8°C), while at TAS3 it was 3.8°C colder (5.1°C vs 8.9°C). Given that heat flux into lakes has been
identified as a key driver of gas production and release in Swedish lakes (Wik et al., 2014), these temperature differences may
have contributed to the observed variations in bubbling activity.

While both intensive campaign periods occurred during the summer season, we acknowledge that they were conducted at
different times, with the 2023 campaign occurring later in the season. Beyond the overall differences in environmental
conditions between both summers–2022 being colder (0.7°C above seasonal normal) than 2023 (2.6°C above seasonal
normal)–the timing of sampling may have captured distinct seasonal phases. Late summer is typically characterized by a
warmer water column and greater accumulation of GHGs at the bottom due to prolonged stratification throughout the season
(Prėskienis et al., 2021).

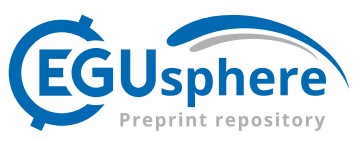

**Table 6. Reported $CO_2$ and $CH_4$ fluxes from thermokarst lakes and ponds. Negative values represent uptake from the atmosphere. Nb is the number of waterbodies studied. Type of permafrost are continuous (C), discontinuous (D) and sporadic (S). Methods for acquiring diffusive flux are chamber ($CH$), headspace equilibration (HS), sensors measuring concentration directly in the field (SR), bubble traps (BT) and wind models (WM). na=not available**

| Site | Reference | Region | Nb | $F_{D\,CO_2}$ | $F_{E\,CO_2}$ | $F_{D\,CH_4}$ | $F_{E\,CH_4}$ | Method |
|---|---|---|---|---|---|---|---|---|
| Nunavik, Canada | This study | D; Lithalsa | 2 | 21.6 (−2.0 − 67.4) | 0.2 (0.0 − 0.3) | 1.5 (0.1 − 7.2) | 9.6 (0.5 − 25.2) | CH, HS, BT |
| | Matveev et al. (2018) | D, S; Lithalsa | 8 | 12.3 (−1.7 − 30.8) | 0.1 (0.006 − 0.2) | 0.4 (0.07 − 0.82) | 0.1 (0.02 − 0.25) | HS, SR, WM, BT |
| | Matveev et al. (2016) | D, S; Palsa | 9 | 58.3 (4 − 242) | 0.01 (0.001 − 0.1) | 3.3 (0.01 − 12.8) | 0.2 (<0.01 − 0.8) | HS, SR, WM, BT |
| Nunavut, Canada | Prėskienis et al. (2021)[1] | C; High Arctic ice-wedge | 21 | 7.1 | 0.1 | 0.4 | 0.9 | HS, WM |
| Northernmost Alberta, Canada | Kuhn et al. (2023) | S; Boreal | 1 | na | 0.54 | 2.56 | 8.21 | BT, HS, WM |
| Tibetan Plateau, Central Asia | Wang et al. (2021) | C, D and S; Alpine | 32 | na | na | 2.6 (0.003 − 48.4) | 6.6 (0.002 − 140.0) | HS, WM, CH |
| | Yang et al. (2023) | D and S; Alpine | 120 | 170.3 (−33.7 − 445.0) | na | 2.1 (0.1 − 7.9) | 11.2 (0.03 − 31.4) | CH, HS |
| Central Yakutia, Eastern Siberia | Hughes-Allen et al. (2021) | C; Peatland | 91 | 19.9 (−9.4 − 355.4) | na | 19.9 (0.1 − 566.4) | na | HS, WM |

[1] Kettle lakes in Prėskienis et al. (2021) were later identified by Coulombe et al. (2022) as glacial thermokarst lakes and thus were included in the average grouping ice-wedge trough ponds, ice-wedge thermokarst lakes, and glacial thermokarst lakes



## 4.4 The importance of diel variability in upscaling emissions

Estimates of GHG emissions from lakes often rely on daytime measurements, but our finding indicate that this approach can significantly overestimate total 24-hour fluxes, as observed by Sieczko et al. (2020) in boreal lakes and Martinez-Cruz et al. (2020) in eutrophic temperate lakes using automated chambers. In 2023, the pronounced diel pattern in surface gas concentrations and $k$ led to significantly higher daytime (9:00-17:00) diffusive fluxes: 47% higher for $CO_2$, 95% for $CH_4$, and 75% for $N_2O$ (negative fluxes for $N_2O$; Fig. 9). This pattern was not observed in July 2022, a colder period characterized by weaker stratification following an intensive mixing period, which likely resulted in gas venting (Fig. 3). The causes of these patterns are multifaceted, involving both physical (mixing patterns) and biological factors (production vs. consumption rates), though the latter were not explored in this study.

Our results suggest that the high morning $CO_2$ fluxes were primarily driven by elevated gas concentrations in the upper water column, illustrated by the significant correlation between flux and departure from saturation ($r = 0.49$, $p = 0.018$), combined with an increase in turbulence ($k$) during the day. The higher morning $CO_2$ concentrations coincided with weaker stratification and lower oxygen levels at that time (Fig. S7d, Fig. 8c,d). This pattern aligns with findings from Arctic thermokarst lakes (Prėskienis et al., 2021), where nighttime respiration and daytime photosynthesis, along with nighttime convective mixing, were suggested to concurrently drive $CO_2$ and $O_2$ dynamics by bringing GHG-enriched, oxygen-depleted water from deeper layers to the surface (Walter Anthony and Macintyre, 2016). Similarly, elevated morning $CO_2$ concentrations have been reported in a boreal humic lake, where surface concentrations were strongly influenced by the stability of stratification (Huotari et al., 2009).

Wind-induced turbulence apparently played a dominant role in controlling fluxes of $CH_4$ and $N_2O$, as these gases were more closely linked to $k$ patterns, with their surface concentrations remaining relatively stable. However, $CH_4$ in TAS3 followed a pattern similar to $CO_2$, highlighting the need for additional measurements to better resolve these diel patterns. Wind-driven dynamics contributing to higher daytime fluxes have also been reported in boreal lakes (Sieczko et al., 2020). These differing dynamics caused $CH_4$ and $N_2O$ fluxes to peak later in the day compared to $CO_2$. This contrasts with findings from Eugster et al. (2020), who reported elevated $CO_2$ and $CH_4$ fluxes at night (2h-6h) in a deep glacial lake, and Podgrajsek et al. (2014), who observed high $CH_4$ fluxes during the night and early morning (00h-08h) in a Swedish lake, both coinciding with periods of strongest surface cooling and convection. These results underscore the importance of conducting full 24-hour sampling to accurately estimate GHG emissions from lakes, as restricting measurements to daytime hours can lead to substantial biases (Sieczko et al., 2020).

## 4.5 Larger-scale estimates and limitations

The larger-scale estimates of GHG fluxes from lakes TAS1 and TAS3 reveal significant seasonal variation in emissions. The lower GHG fluxes recorded during the summer of 2022 were primarily due to weaker stratification and lower surface water temperatures, which limited the accumulation of GHGs in the water column. The larger GHG fluxes observed during the



summer of 2023 reflects enhanced stratification and intensified decomposition processes, leading to greater accumulation and subsequent release of $CO_2$ and $CH_4$.

The higher fluxes estimated at TAS3 in the fall and spring can be attributed to the increased accumulation of GHGs in the

deeper layers during the summer and winter months. While we were unable to estimate $k$ during the fall 2023 and spring 2024 periods, it is likely that fluxes would have been substantially higher. Our estimates suggest that surface dissolved gas concentrations during these periods were 3.8 times higher for $CO_2$ and 7.8 times higher for $CH_4$ compared to fall 2022 and spring 2023 (Table S6). Overall, these findings highlight the importance of monitoring GHG fluxes year-round to gain a deeper understanding of the dynamics and drivers behind seasonal emission patterns.

Several factors should be considered to improve the accuracy of GHG flux estimates. First, in cases of strong stratification (as in 2023), gas concentration gradients may be pronounced even near the surface. When quantifying gas concentration with the headspace method, water is typically collected from the upper 10-20 cm, which could overestimate concentrations at the very surface, especially during warm, calm periods, potentially inflating flux estimates during the day when applying a gas transfer model. Conversly, all measurements in the present study were taken at the lake centre, which could result in an underestimation

of fluxes. Previous studies, such as Kuhn et al. (2023), have shown that fluxes near the edges can be significantly higher (twice in their study) than at the centre. Finally, with only one funnel used per lake to measure ebullition (despite the lakes being very small), we may have missed localized high-ebullition sites, particularly near the edges. For instance, Kuhn et al. (2023) reported $CH_4$ ebullition fluxes up to four times higher at the thawing edges of thermokarst lakes compared to the lake centre.

## 5 Conclusions

In this study, two thermokarst lakes–one more humic and sheltered and the other more transparent and wind-exposed–were monitored for nearly two years, with continuous measurements of temperature, oxygen, and conductivity. These long-term observations were complemented by GHG flux and concentration measurements from two intensive summer campaigns–one colder and the other warmer–to quantify emissions and their variability. The thermokarst lakes were significant emitters of $CO_2$ and $CH_4$, with lake $N_2O$ uptake only minimally offsetting these emissions. Our results show large temporal variations in

GHG fluxes during the warm intensive campaign in 2023, with peak $CO_2$ fluxes occurring in the morning and peak $CH_4$ and $N_2O$ fluxes observed later in the day. The differences observed between the warmer and the cooler intensive campaigns were partly due to the mixing regime and the accumulation of dissolved gases in the water column. Stratification allowed for the buildup of dissolved GHGs, which were then brought to the surface during nighttime cooling and mixing, subsequently released to the atmosphere as wind speeds rose. Notably, ebullitive $CH_4$ fluxes were predominant in the more humic and

sheltered lake, which receives organic inputs from leafy shrubs, illustrating the potential differences between two adjacent lakes. Our seasonal flux estimates revealed that emissions should be highest in fall and spring, with $CH_4$ contributing the majority of the total flux, highlighting the role of GHG storage in stratified water bodies. This pronounced temporal variability



underscores the importance of long-term monitoring that accounts for both seasonal and diel patterns in emissions, alongside the environmental factors that regulate the efficiency of lake mixing and GHG production and consumption.


**Data availability.** The datasets generated and/or analysed during this study will be available on Borealis, the Canadian Dataverse Repository at https://doi.org/10.5683/SP3/KCX9KV.

**Author contributions.** IL and AP planned the campaign; AT installed the buoy and sensor system; AP, IL and AT conducted
the measurements; AP analysed the data; AP wrote the manuscript, with all authors providing feedback on earlier drafts and approving the final version.

**Competing interests.** The contact author has declared that none of the authors has any competing interests.

**Acknowledgements.** We would like to express our sincere gratitude to the Inuit community of Umiujaq for granting permission to work in the study area. Our thank also go to Jean-Michel Lemieux for his invaluable insights into the evolution of thermokarst lakes in the Tasiapik Valley, as well as to Denis Sarrazin and Florent Dominé for providing key data for this study. We are also grateful to Geneviève Beaudoin and Clarence Gagnon for their essential field assistance.

**Financial support.** This project was funded by the Université Laval Sentinel North program (DN) and the NSERC Discovery grant (IL), with additional logistical support from the Centre d'Études Nordiques (CEN).



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
