# Peer review of "Temporal patterns of greenhouse gas emissions from two small thermokarst lakes in Nunavik, Canada"

_EGUsphere, 2025_

## Author Comment (AC1)

**Temporal patterns of greenhouse gas emissions from two small thermokarst lakes in Nunavik, Canada**

**Responses to Dr. Matthias Koschorreck**

We would like to thank Dr. Matthias Koschorreck for his helpful contribution to the article. Please note that suggested additions to the article are shown in **bold**. The lines in this document refer to the previous version of the manuscript and may be subject to change in the revised version.

1.      General remarks

The manuscript reports results from a) ca. 1.5 year continuous measurements of limnophysical and meteorological data and b) GHG data from 2 short and intensive summer campaigns in two Canadian thermokarst lakes. From these data the authors try to figure out how GHG emissions in the two lakes were regulated and come up with seasonal GHG budgets for the two lakes.

The topic of the study is important and interesting. GHG emissions from thermokarst lakes are relevant and not well studied. This is probably partly due to the logistic challenges connected with the topic. This makes the presented data very valuable. I appreciate the amount and quality of the dataset obtained under challenging logistic constraints.

Thank you, your kind words are greatly appreciated.

2.      Major concerns

2.1.    In their third hypothesis they state that the study was conducted to estimate seasonal variability. However, the method approach is not suited to address seasonal variability of GHGs. Only measuring GHG concentrations and fluxes during two short summer campaigns does not allow to study seasonal dynamics. The authors try to circumvent the lack of seasonal GHG data by a clever combination of assumptions and limnophysical data. This is feasible but cannot be a core element of the paper. Seasonal conclusions are not sufficiently supported by data. I recommend to adjust the aim of the paper and tone down the seasonal part of the paper accordingly. I recommend to focus more on the strong aspects of the paper: The combination of limnophyscial, meteorological and GHG data during the campaigns as well was the seasonal pattern of stratification and oxygen.

Thank you for this important comment. We agree that our dataset does not allow a full quantification of seasonal GHG variability. In view of this, we propose these revised objectives to clarify that the focus of the study is on understanding the drivers of summer temporal variations in GHG fluxes by measuring them and emphasize the exploration of seasonal patterns using limnophysical proxies, rather than directly measuring seasonal GHG emissions:

Lines 91 : The objectives were to (1) characterise their physicochemical properties, (2) assess the magnitude and diel variation of summer GHG fluxes, and (3) **explore how seasonal patterns in limnophysical conditions may influence GHG dynamics.**

We also suggest moving the Methodology and Results sections on seasonal extrapolations, including Table 5, to the supplementary materials (as Table S7). Consequently, we would only discuss these extrapolations in the discussion section (4.5 Larger-scale estimates and limitations), which has also been modified to improve clarity. These major changes could shorten the Methodology section by 23% and the Results section by 11%. Finally, we propose toning down the conclusions regarding seasonal patterns, presenting them now as hypothesis-generating interpretations rather than firm conclusions:

Lines 681–684: **While** our seasonal flux estimates **suggest** that emissions **may** peak in fall and spring, highlighting the role of GHG storage and release in stratified water bodies, **these remain speculative, as no direct seasonal GHG measurements were conducted.** This highlights the importance of long-term monitoring that accounts for both seasonal and diel patterns in emissions, alongside the environmental factors that regulate the efficiency of lake mixing and GHG production and consumption.

2.2.    The manuscript is rather long. This is partly because there are several redundancies between text and figures and tables (e.g. l.443, l.449) and also redundancy between discussion and results (see below). Also, the discussion is in large parts a repetition of the results and does not contain much new information. This is particularly true for section 4.1. Also l.581-584 or l.590.

Thank you for this helpful observation. To reduce redundancy between the text, figures, and tables, we removed Table 4 from the main text and transferred it to the supplementary material, as you suggested in Comment 3.23. We also shortened Section 3.1 by 11% by removing overly detailed passages that were already clearly conveyed by the figures.

To address redundancies between the Results and Discussion sections, we substantially revised Section 4.1 (Physicochemistry), focusing more on interpretation rather than description, and moved some descriptions to Section 3.3 (Vertical structure of the water column during the intensive sampling periods). We also revised Section 4.3 (Summer GHG fluxes) to reduce repetition. These changes shortened the Discussion section by 9%.

Finally, combining all the comments, the proposed changes contributed to streamlining the manuscript by 8.3%, reducing the word count from 14,573 to 13,369. These modifications have improved the text's clarity and conciseness, while retaining the necessary scientific content.

2.3. A strength of the dataset is that you not only calculated but also measured k600 (what not many people do). You hide in the supplement that both methods agreed fairly well. I recommend to exploit this more in the manuscript. Another chance you missed is: You quantified fluxes from k and concentration. Thus, you can exactly quantify the role of k versus concentration for flux dynamics (and not just write "likely" as in line 593 or "suggest" in l.629). See e.g.

https://aslopubs.onlinelibrary.wiley.com/doi/10.1002/lno.12528.

Thank you for highlighting this strength and for the helpful reference. We acknowledge that the agreement between measured and modelled $k_{600}$ values is an important point of validation. However, the $k_{600}$ values presented in Table S5 (now updated as Table S8) are seasonal estimates based on the limnological seasons, rather than direct point-by-point comparisons during the intensive summer campaigns. We suggest adding a table of the point-by-point values of $k_{600}$ to Supplementary Material:

Table S6. Gas transfer velocities ($k_{600}$) measured with the floating chamber in lakes TAS1 and TAS3 using surface GHG concentrations obtained by the headspace method. Time of day is expressed in EST.

| Lake | July 2022 | | August 2023 | |
| | Date | $k_{600}$ (cm h$^{-1}$) | Date | $k_{600}$ (cm h$^{-1}$) |
| --- | --- | --- | --- | --- |
| TAS1 | 2022-07-07 09:14 | 4.00 | 2023-08-10 11:50 | 10.12 |
| | 2022-07-07 13:36 | 4.18 | 2023-08-11 19:10 | 9.80 |
| | 2022-07-11 01:59 | 4.51 | 2023-08-12 22:41 | 3.19 |
| | 2022-07-11 16:09 | 4.97 | 2023-08-13 18:18 | 4.88 |
| | 2022-07-12 11:53 | 4.08 | 2023-08-14 10:28 | 5.76 |
| | 2022-07-12 19:03 | 2.54 | 2023-08-14 13:48 | 11.01 |
| | 2022-07-13 08:37 | 1.57 | 2023-08-15 01:23 | 1.61 |
| | 2022-07-13 10:51 | 4.29 | 2023-08-15 16:55 | 4.46 |
| | 2022-07-15 04:19 | 1.62 | 2023-08-16 05:09 | 4.37 |
| | | | 2023-08-18 11:41 | 4.13 |
| | | | 2023-08-19 08:00 | 3.09 |
| | | | 2023-08-20 08:00 | 5.36 |
| TAS3 | 2022-07-07 10:41 | 2.89 | 2023-08-11 20:19 | 2.38 |
| | 2022-07-07 14:22 | 1.96 | 2023-08-12 23:36 | 1.60 |
| | 2022-07-08 08:42 | 1.78 | 2023-08-13 19:42 | 3.03 |
| | 2022-07-11 00:59 | 2.57 | 2023-08-14 11:45 | 4.52 |
| | 2022-07-12 11:11 | 1.17 | 2023-08-14 14:42 | 8.19 |
| | 2022-07-12 19:47 | 1.36 | 2023-08-15 02:15 | 0.45 |
| | 2022-07-13 09:50 | 1.48 | 2023-08-15 17:53 | 2.98 |
| | | | 2023-08-16 07:18 | 2.54 |
| | | | 2023-08-18 12:46 | 1.88 |
| | | | 2023-08-19 08:54 | 1.33 |
| | | | 2023-08-20 09:03 | 3.43 |

For clarification, we also propose revising the captions of Tables S6 and S8 (now Tables S7 and S9) in the Supplementary Material to better reflect their temporal scope:

> Table S6. Seasonal diffusive flux estimates at lakes TAS1 and TAS3 **across limnological seasons. For summer comparisons, the period from 7 July to 20 August was used; fall 2022 and spring 2023 correspond to September-October and mid-May to mid-June, respectively.** Values are expressed as min - max for each period. na = not available.

> Table S8. Gas transfer velocities ($k_{600}$) estimated with the surfacer renewal model in lakes TAS1 and TAS3 **across limnological seasons. For summer comparisons, the period from 7 July to 20 August was used; fall 2022 and spring 2023 correspond to September-October and mid-May to mid-June, respectively.** Values are expressed as mean (min-max).

In response to your second point, we agree that our dataset offers a valuable opportunity to disentangle the respective roles of $k$ and dissolved gases in controlling flux dynamics. To address this, we fitted sine curves to the diel cycles of fluxes, gas concentrations, and $k$ values to better visualize temporal patterns (Figure 9). We suggest the following modifications to be more transparent in the Results section:

> Line 446: Average diffusive GHG emissions from TAS1 were five times higher than those from TAS3, consistent with the higher surface concentrations observed at TAS1 **as well as the more turbulent conditions (average $k_{600}$ of 3.5 cm h$^{-1}$ at TAS1 and 1.9 cm h$^{-1}$ at TAS3) (Table S6). One of our $k$ estimates was negative (July 11 2022, 17:15 at TAS3; see Table S2), which can happen near equilibrium due to uncertainties in the measurements. This value was excluded from the analysis.**

> Line 452: Despite total diffusive fluxes being similar between TAS1 and TAS3 in 2023, surface concentrations of $CO_2$ and $CH_4$ were lower at TAS1 (Table 3). This difference is due to the more turbulent conditions at TAS1 **(average $k_{600}$ of 5.7 cm h$^{-1}$),** which increased the overall emissions **compared to those at TAS3 (average $k_{600}$ of 2.9 cm h$^{-1}$; see Table S6).**

We then assessed the relative importance of $k$ and dissolved gases in explaining flux variability, calculating their correlation with fluxes over the diel cycle. We propose including this analysis in the Results section, adding the following sentence at line 638 of the revised manuscript:

> Line 638: Wind-induced turbulence played a dominant role in controlling $CH_4$ and $N_2O$ fluxes, as these gases were more strongly correlated with $k$ **($r = 0.78$, $p < 0.001$ for $CH_4$; $r = -0.98$, $p < 0.001$ for $N_2O$),** while their surface concentrations remained relatively stable.

While a full dominance analysis would be ideal, our current sample size (9–12 points per diel cycle) limits the complexity of the statistical tools we can apply. Nonetheless, the added curve fitting and correlation analysis address your comment and better highlight the mechanistic drivers of diel flux variability.

[Figure]

**Figure 9.** Diel cycles of $CO_2$ (a,d), $CH_4$ (b,e) and $N_2O$ (c,f) diffusive fluxes for the July 2022 (left) and August 2023 (right) intensive summer campaign periods. $CO_2$ fluxes were measured directly with the floating chamber, while $CH_4$ and $N_2O$ fluxes were obtained by applying the $k$ values derived from $CO_2$ fluxes to their respective concentrations. The grey dotted line represents atmospheric equilibrium, **and the grey shaded area indicates periods of zero solar radiation. Although the data are presented on a diurnal cycle, the measurements were taken over a two-week period (see Table S2). Coloured dotted lines represent sine-fitted trend lines.** Time of day is expressed in EST.

[Figure]

Figure S7. **Diel cycles of** near-surface concentrations of $CO_2$ (a,d), $CH_4$ (b,e) and $N_2O$ (c,f) for the July 2022 (left) and August 2023 (right) intensive summer campaign periods for TAS1 and TAS3. **The grey shaded areas indicate periods of zero solar radiation. Although the data are presented on a diurnal cycle, the measurements were taken over a two-week period (see Table S2). Coloured dotted lines represent sine-fitted trend lines.** Time of day is expressed in EST.

[Figure]

Figure S8. **Diel cycles of** gas transfer coefficient estimated from the gas chamber method in (a) July 2022 and (b) August 2023 for lakes TAS1 and TAS3. **The grey shaded areas indicate periods of zero solar radiation. Although the data are presented on a diurnal cycle, the measurements were taken over a two-week period (see Table S2). Coloured dotted lines represent sine-fitted trend lines.** Time of day is expressed in EST.

3. Detailed remarks

3.1. l.13: It's a bit confusing that a colder summer had temperatures above long term average. Isn´t it more a hot versus a warm summer?

Thank you for pointing out this ambiguity. The intention was to refer specifically to the fieldwork periods rather than the entire summer seasons. In July 2022, during the first campaign, the mean temperature was considerably lower than in August 2023 (8.8°C vs. 14.6°C), despite being slightly above the long-term June–August mean. To avoid confusion, we have reworded this sentence to clarify that we are comparing campaign periods rather than seasonal means:

Line 13: One campaign occurred during a colder **period (8.8°C average temperature)** and the other during a warmer one (**14.6°C average temperature**).

To further reduce confusion, we adjusted the seasonal comparison to focus on mid-June to the end of August (instead of June 1st onward), as early June often reflects spring conditions at this latitude. We also clarified that the campaign conditions are now compared to the corresponding monthly normal rather than the broader seasonal average:

Line 358: Overall, summer 2022 was colder than summer 2023. The average air temperature **from mid-June to August** was **10.3 ± 4.7°C in 2022, compared to 13.6 ± 4.3°C in 2023. In 2022, the average temperature over this period was 0.9°C lower than normal, whereas in 2023 it was 2.4°C higher than normal**

Line 367: Weather conditions varied considerably between the two intensive campaign periods, with July 2022 being significantly colder and windier than August 2023. **During the July 2022 campaign**, the mean air temperature was 8.8 °C (range: 1.3–23.1 °C), **which is 2.7 °C below the July normal of 11.5 ± 5.1 °C. In**

**contrast,** August 2023 was substantially warmer, **with an average temperature of** 14.6 °C (range: 6.6–23.4 °C), **which is +2.8 °C above the August normal of 11.8 ± 4.1 °C.**

3.2.    l.52.: "… ebullitive CH4 fluxes…"

This is indeed the case; it will be corrected.

3.3.    l.120: They were not permanently anoxic in the bottom water. Maybe change to "with frequent anoxic conditions"

Thank you, it will be corrected.

3.4.    l.125: I doubt that Secchi depth can be measured at 1 cm precision. Remove "very".

Thank you, we will adjust the reported values to 1.20 m for TAS1 and 0.55 cm for TAS3 in Section 2.2 and in Table 1. The word "very" will also be removed from line 125.

3.5.    l.202: "Did you really deploy only one bubble trap per lake? As it becomes clear later in line 666 they indeed only deployed one trap. This is in my eyes not enough to come up with a robust estimate of ebullition.

Yes, we only deployed one bubble trap per lake. We acknowledge the limitations this imposes on the spatial representativity of our ebullition estimates. However, due to logistical constraints and the small size of the lakes (with diameters of about 10 m), installing multiple traps was not feasible. To make this clearer, we have added a note to the methodology.

Line 202: Ebullitive fluxes of $CO_2$ and $CH_4$ were measured **at the center of the lake** using an inverted funnel submerged below the water surface.

We also address this limitation in the discussion, emphasizing the need for caution in generalizing ebullitive flux rates from these data and suggesting directions for future work:

Line 605: **However, we acknowledge that our estimates of $CH_4$ ebullition were based on measurements taken from a single funnel deployed at the center of the lake. This approach restricts the spatial representativeness of the results, especially since ebullition is known to be highly heterogeneous, with localized hotspots often occurring near lake edges or thawing permafrost zones (Wik et al., 2011; Kuhn et al., 2023). This spatial limitation may partially explain the differences observed between the lakes. Future studies should therefore aim to deploy multiple bubble traps across various lake zones to capture this heterogeneity and better constrain whole-lake ebullition estimates.**

3.6.   l.215.: How were the exetainers vacuumed?

Thank you for your question. The Exetainer vials were vacuumed twice for 3 minutes using a vacuum pump, with a 10-15 second nitrogen flush in between. To verify the vacuum, we tested one vial every 10 to 20 by submerging it in water, piercing it with a needle, and checking whether the vial quickly filled with water, leaving only a small residual air bubble. This procedure ensured adequate vacuuming for gas sampling. We propose clarifying this in the manuscript:

> Line 213: The bottle was shaken vigorously for 3 min, and 20 mL of the gaseous headspace was transferred to a 12 mL gas-tight Exetainer vial that had been pre-flushed **twice for 10-15 s** with nitrogen and vacuumed **twice for 3 min**.

3.7.   l.223.: It is not very common to use a (not very sensitive) TCD for this kind of study. This limits the precision of the concentration data. Please report detection limits and analytical precision for dissolved gas concentrations – not only ppm results.

Thank you for your comment. The GC used for our analyses is equipped with 4 detectors (FID X2, TCD and ECD). We used 2 channels into which 5 mL of gas sample is injected and separated in 2 different injection loops. The first, which is fitted with a TCD and FID detector in series, was used to analyse $CH_4$ and $CO_2$. More specifically, the TCD (non-destructive and less sensitive analysis) doses high concentrations, after which the analyte is directed to the FID, which doses lower concentrations. The second, fitted with an ECD detector (which is highly sensitive) analyses nitrous oxide ($N_2O$) by electron capture. We propose the following clarification in the manuscript:

> Line 216: Gas samples were analysed using a Trace 1310 gas chromatograph (Thermo Fisher Scientific). **We used two parallel channels, one with TCD-FID detectors (column HSQ 80/100, MS 5A 80/100) for $CH_4$ and $CO_2$, and one with an ECD detector (column HSQ 80/100) for $N_2O$.** Calibration curves were established for $CO_2$ (up to 10 000 ppm), $CH_4$ (up to 45 000 ppm), and $N_2O$ (up to 1 ppm). Detection limits were 200 ppm for $CO_2$, 3 ppm for low $CH_4$ concentrations, 50 ppm for $CH_4$ concentrations above 1000 ppm, and 0.1 ppm for N2O.

3.8.   l.225: Was the CO2 change during the chamber measurements linear? 30 min is a rather long time for these measurements. Can you prove that linear fitting is not under-estimating fluxes?

Yes, the $CO_2$ change during the chamber measurements was linear. The coefficient of determination ($R^2$) values from the linear regressions were consistently high, with average $R^2$ values of 0.94 in 2022 and 0.99 in 2023. We suggest clarifying this in the Result section:

> Line 443: In 2022, $CO_2$ diffusive fluxes ranged from –2.0 to 17.1 mmol m$^{-2}$ d$^{-1}$, **with $R^2$ values from the linear regressions averaging 0.94.**

Line 449: In 2023, CO₂ diffusive fluxes ranged from 7.8 to 67.4 mmol m⁻² d⁻¹, **with R² values from the linear regressions averaging 0.99.**

3.9. l.227: Was the inner volume of the analyzer and the tubes involved?

Thank you for your comment. Yes, the volume of the tube was included, but not the analyzer. We will clarify it at line 224.

3.10. l.242: Maybe add position of the traps to figure 1.

Thank you for the suggestion. We have now added the location of the traps to Figure 1, along with a corresponding note in the caption for clarity. We also suggest adding the location of the mooring lines.

[Figure]

Figure 1. Location map of the Tasiapik Valley: (a) Map of Canada highlighting the study site marked with a red dot, and (b) areal contours of the two lakes under study. **The locations of the mooring lines are marked with red stars, while the bubble traps are marked with yellow crosses.** Aerial photography by Madeleine St-Cyr.

3.11. l.247: A partial pressure is not dimensionless but has a pressure unit. What you probably mean is "volumetric mixing ratio [ppmv]"?

Thank you for your comment, you are right and will be corrected (see below).

3.12. l.249: How was the molar volume obtained?

We calculated the molar volume using the ideal gas law, based on measured air temperature and pressure at the site. This will be clarified in the revised text:

> Line 248: where $P$ is the partial pressure of the gas (**ppmv**), $V_g$ is the total volume of gas in the funnel (m³), $\Delta t$ is the time interval between two measurements (days), $V_m$ is the molar volume of gas at local air temperature **and pressure, calculated using the ideal gas law** (m³ mol⁻¹), and $A$ is the collection area of the funnel (m²).

3.13. l.300…: use past tense for results.

Thank you. We will correct the verb tenses in the Results section to consistently use the past tense.

3.14. l.334: I do not see a data gap in the figure.

Thank you for your comment. We assume you are referring to the red dotted line in Fig. 3b, which indicates the discarded data from 26 June to 14 July 2023. To avoid confusion, we suggest removing the line and leaving a gap instead. We have also adjusted the "ice cover" box to fit 10 November 2022 as it was on 1 November 2022 by mistake.

[Figure]

Figure 3. Physicochemical profiles from summer 2022 to summer 2023, showing temperature and buoyancy frequency *N* for (a) TAS1 and (c) TAS3, with the water surface marked by a black dotted line. Oxygen saturation and specific conductivity (SC) are shown for (b) TAS1 and (d) TAS3. Temperature at the very surface was not measured, as the buoy was positioned below the air-water interface to prevent it from becoming trapped in the ice cover. Nevertheless, winter temperature data suggest that ice may have grown over the buoys, pushing down the mooring and changing the depth of surface sensors, making surface data unreliable. Intensive summer campaign periods are indicated by brackets for July 2022 and August 2023.

3.15.  l.375: It is not very clear what "vertical structure" means. Maybe "vertical structure of the water column"?

Thank you for pointing this out. We will adjust the section title as follows:

Line 375: 3.3 Vertical structure **of the water column** during the intensive sampling periods

3.16.  l.376-77: Remove sentence.

Thank you, it will be removed.

3.17.  l.392: Reformulate "While surface remained similar".

The sentence is indeed unclear; we propose this reformulation:

Line 392: Additionally, the specific conductivity **increased considerably with depth**, particularly at TAS1 (up to 128 µS cm$^{-1}$), which contributed to stronger density stratification.

3.18.  l.393: Can you calculate how the density effect of the conductivity change compares to the temperature induced density?

Thank you for your suggestion. We have now included a quantitative comparison between the effects of temperature and conductivity on water density. Specifically, we calculated the density contribution of conductivity using the simplified empirical relationship $\rho = 1000 + 0.000695\text{TSD}$ kg m$^{-3}$ (Collins, 1987).

Line 393: Additionally, the specific conductivity increased considerably with depth, particularly at TAS1 (up to 128 µS cm$^{-1}$ **or total dissolved solids of 123 mg L$^{-1}$**), which contributed to stronger density stratification. **For example, temperature accounted for a density difference of 1.796 kg m$^{-3}$ on 13 August while conductivity contributed approximately to 0.074 kg m$^{-3}$, based on the simplified relationship of $\rho = 1000 + 0.000695\text{TSD}$ kg m$^{-3}$ (Collins, 1987).**

3.19.  l.396: What do you mean by "tilting"? Tilting of the thermocline?

Yes, we use "tilting" referring to the displacement of the thermocline due to wind forcing. To clarify this, we will reformulate the sentence as follows:

Line 395-397: During the 2022 intensive campaign, $W$ mostly indicated partial **thermocline** tilting ($1 < W < 10$) during the day at TAS3, whereas at TAS1, they generally suggested minimal tilting ($> 10$).

3.20. l.426: Change to "atmospheric equilibrium values".

Thank you, this will be corrected.

3.21. l.454: Why only "likely". You quantified k, so you should know.

Thank you for your comment. "Likely" will be removed, along with the updates related to *k* analysis made in response to comment 2.3.

3.22. L458: Please report bubble CH4 content data.

Thank you for your comment. We have now included the exact bubble content data in the dataset available on Borealis. This information has also been specified in the revised manuscript at line 456.

3.23. Table 4 is redundant to Figure 7. The table can be removed to the supplement.

Thank you for your comment. Table 4 will be removed to the supplements.

3.24. l.479: It looks as if concentrations were lowest 3 h after sunrise. Any explanation for this lag?

Thank you for this observation. The apparent lag in minimum surface oxygen concentrations ~3 h after sunrise may result from a combination of physical and biological processes. First, nighttime biological oxygen demand (e.g., respiration) continues through the early morning, and may exceed photosynthetic production until sufficient light levels are reached. Second, increasing wind speeds after sunrise could induce vertical mixing, redistributing oxygen-depleted water from the lower layers to the surface. This mixing could contribute to the observed decline in surface oxygen, despite rising irradiance. We have added this interpretation to the discussion (line 628) to clarify this pattern.

3.25. l.486: Why should concentration change when temperature changes. If you express concentration as %sat then yes. But absolute concentrations not. Maybe it makes more sense to use absolute concentration rather than saturation in Figure 8?

Thank you for your comment. We initially presented % saturation in Figure 8 to emphasize the oxygen oversaturation observed during the day (which indicates net photosynthesis), as well as undersaturation at night (due to respiration and/or physical processes). However, we recognize that using absolute concentrations may better isolate biological and physical drivers without conflating them with temperature effects on solubility. We agree to modify Figure 8 to show absolute oxygen concentrations instead and clarifying this change in the caption.

[Figure]

Figure 8. Diel cycles of buoyancy frequency N (a,c) and surface dissolved oxygen at depth z (c,d) for the July 2022 (left) and August 2023 (right) intensive summer campaign periods. The colored shaded area represents the standard deviation **calculated for each hour of the day based on all available measurements within that hour.** The grey shaded area indicates periods of zero solar radiation. Time of day is expressed in EST.

3.26. l.496: How was the standard deviation calculated? Figure: Would be nice to have lines with wind speed or k in the plots.

Thank you for your question. The standard deviation was calculated for each hour of the day based on all available measurements within that hour across the campaign period. This approach reflects the variability observed at each hour throughout the diel cycle. We have clarified this in the figure title at line 496 (see comment 3.25).

Regarding the addition of wind speed or $k$ to the plots, we appreciate the idea but chose not to include them in order to maintain visual clarity and avoid overloading the main figure. Instead, diel variations in $k$ are presented in Supplementary Figure S8.

3.27. l.560: is "heightend" the right wording?

Thank you for your comment. We propose the following reformulation:

Line 559 – 561: The increased bottom concentrations of $CO_2$ and $CH_4$ in 2023 may also reflect **enhanced** microbial activity and **increased** organic matter decomposition in the hypolimnion driven by higher temperatures (see below).

3.28. l.595: How can this be done? Sample more lakes? Sample more site within a lake?

Thank you for this insightful question. We agree that clarifying how spatial variability could be better accounted for would strengthen the statement. We have revised the sentence to reflect this:

Lines 593–595: Thus, GHG dynamics can differ significantly between nearby lakes. Large-scale estimates and comparisons should account for these spatial differences **by including a broader range of lake types and multiple sampling times.**

3.29. l.614: The temperature data are redundant and not needed here.

Thank you, it will be removed.

3.30. l.622: There are several other studies reporting diurnal pattern (e.g Golub, M., et al. (2023). "Diel, Seasonal, and Inter-annual variation in carbon dioxide effluxes from lakes and reservoirs." Environmental Research Letters and references therein)

Thank you for the suggestion. We agree that several recent studies, including that of Golub et al. (2023), have demonstrated the importance of considering diel variability in lake GHG fluxes. We will revise the sentence to better acknowledge the literature on this topic:

Line 622: Estimates of GHG emissions from lakes often rely on daytime measurements, but our findings indicate that this approach can significantly overestimate total 24-hour fluxes. **Numerous studies have also reported diel variability in subtropical lakes (Zhao et al., 2024), boreal and temperate lakes (**Sieczko et al., 2020**; Erkkilä et al., 2018; Podgrajsek et al., 2014;** Martinez-Cruz et al., 2020**), subarctic lakes (Jammet et al., 2017), and glacial lakes (Eugster et al., 2022), as well as across multiple sites in the Northern Hemisphere (Golub et al., 2023).**

3.31. l.647-649: This is not really a new conclusion. Maybe focus more on the special case of thermokarst lakes in the sense "not only in eutrophic warm lakes with a lot of biology but also in this peculiar thermokarst lakes diurnality needs to be considered". Another interesting aspect of diurnality in high latitudes could be that (different to "normal" settings) there are periods without darkness in summer.

Thank you for your comment. We agree that the focus should be placed more explicitly on the unique case of thermokarst lakes. We propose the following revised passage to clarify this point:

Lines 647: These results highlight the importance of conducting full 24-hour sampling in order to accurately capture GHG emissions from lakes, **not only in biologically productive systems, but also in high-latitude thermokarst environments. Despite their small size and shallow depth, these lakes exhibit pronounced diel patterns in stratification and oxygen dynamics, demonstrating that GHG fluxes are largely driven by physical processes. Furthermore, the distinctive light regimes of the Arctic and sub-Arctic regions, characterised by prolonged daylight hours, could intensify stratification during the summer. This emphasises the importance of considering latitude-specific dynamics when evaluating lake GHG budgets.**

3.32. l.661: See also https://bg.copernicus.org/articles/22/1697/2025/ and Table S2: Why not add the chamber flux data to this table?

Thank you for your comment. We will add the chamber flux data to Table S2 as suggested. We also appreciate you sharing the reference on surface $CO_2$ gradients and have now included it in the manuscript.

Line 661: First, **under calm conditions, microstratification can slow the vertical exchange of gases even within the top few centimeters of the water column. Aurich et al. (2025) demonstrated that $CO_2$ concentrations may vary significantly between 5 and 25 cm depth during windless periods, with the surface being supersaturated**. When quantifying gas concentration with the headspace method, water is typically collected from the upper 10-20 cm, which could overestimate concentrations at the very surface during warm, calm periods, potentially inflating flux estimates when applying a gas transfer model.

**References**

Aurich, P., Spank, U., and Koschorreck, M.: Surface $CO_2$ gradients challenge conventional $CO_2$ emission quantification in lentic water bodies under calm conditions, Biogeosciences, 22, 1697-1709, 10.5194/bg-22-1697-2025, 2025.

Collins, A.: Properties of produced waters, in: Petroleum engineering handbook, Society of Petroleum Engineers, Richardson, Texas, 1987.

Erkkilä, K. M., Ojala, A., Bastviken, D., Biermann, T., Heiskanen, J. J., Lindroth, A., Peltola, O., Rantakari, M., Vesala, T., and Mammarella, I.: Methane and carbon dioxide fluxes over a lake: comparison between eddy covariance, floating chambers and boundary layer method, Biogeosciences, 15, 429-445, 10.5194/bg-15-429-2018, 2018.

Eugster, W., DelSontro, T., Laundre, J. A., Dobkowski, J., Shaver, G. R., and Kling, G. W.: Effects of long-term climate trends on the methane and $CO_2$ exchange processes of Toolik Lake, Alaska, Front. Env. Sci., 10, 10.3389/fenvs.2022.948529, 2022.

Golub, M., Koupaei-Abyazani, N., Vesala, T., Mammarella, I., Ojala, A., Bohrer, G., Weyhenmeyer, G. A., Blanken, P. D., Eugster, W., Koebsch, F., Chen, J., Czajkowski, K., Deshmukh, C., Guérin, F., Heiskanen, J., Humphreys, E., Jonsson, A., Karlsson, J., Kling, G., Lee, X., Liu, H., Lohila, A., Lundin, E., Morin, T., Podgrajsek, E., Provenzale, M., Rutgersson, A., Sachs, T., Sahlée, E., Serça, D., Shao, C., Spence, C., Strachan, I. B., Xiao, W., and Desai, A. R.: Diel, seasonal, and inter-annual variation in carbon dioxide effluxes from lakes and reservoirs, Environm. Res. Lett., 18, 034046, 10.1088/1748-9326/acb834, 2023.

Jammet, M., Dengel, S., Kettner, E., Parmentier, F. J. W., Wik, M., Crill, P., and Friborg, T.: Year-round $CH_4$ and $CO_2$ flux dynamics in two contrasting freshwater ecosystems of the subarctic, Biogeosciences, 14, 5189-5216, 10.5194/bg-14-5189-2017, 2017.

Kuhn, M. A., Schmidt, M., Heffernan, L., Stührenberg, J., Knorr, K.-H., Estop-Aragonés, C., Broder, T., Gonzalez Moguel, R., Douglas, P. M. J., and Olefeldt, D.: High ebullitive, millennial-aged greenhouse gas emissions from thermokarst expansion of peatland lakes in boreal western Canada, Limnol. Oceanogr., 68, 498-513, 10.1002/lno.12288, 2023.

Martinez-Cruz, K., Sepulveda-Jauregui, A., Greene, S., Fuchs, A., Rodriguez, M., Pansch, N., Gonsiorczyk, T., and Casper, P.: Diel variation of $CH_4$ and $CO_2$ dynamics in two contrasting temperate lakes, Inland Waters, 10, 333-347, 10.1080/20442041.2020.1728178, 2020.

Podgrajsek, E., Sahlée, E., and Rutgersson, A.: Diurnal cycle of lake methane flux, J. Geophys. Res.-Biogeosciences, 119, 236-248, 10.1002/2013JG002327, 2014.

Sieczko, A. K., Duc, N. T., Schenk, J., Pajala, G., Rudberg, D., Sawakuchi, H. O., and Bastviken, D.: Diel variability of methane emissions from lakes, P. Natl. A. Sci., 117, 21488-21494, 10.1073/pnas.2006024117, 2020.

Wik, M., Crill, P. M., Bastviken, D., Danielsson, Å., and Norbäck, E.: Bubbles trapped in arctic lake ice: Potential implications for methane emissions, J. Geophys. Res.-Biogeosciences, 116, G03044, 10.1029/2011JG001761, 2011.

Zhao, F., Huang, Z., Wang, Q., Wang, X., Wang, Y., Zhang, Q., He, W., and Tong, Y.: Seasonal pattern of diel variability of $CO_2$ efflux from a large eutrophic lake, J. Hydrol., 645, 132259, 10.1016/j.jhydrol.2024.132259, 2024.

---

## Author Comment (AC2)

**Temporal patterns of greenhouse gas emissions from two small thermokarst lakes in Nunavik, Canada**

**Responses to Anonymous Referee #2**

We would like to thank the reviewer for their helpful contribution to the article. Please note that additions to the article are shown in **bold**. The lines in this document refer to the previous version of the manuscript and may be subject to change in the revised version.

1.    General remarks

Here, Pouliot et al. conducted an intensive study on GHG emissions in two thermokarst lakes located within the same ecosystem but exhibiting contrasting limnological characteristics. The study primarily aims to synthesize data from two consecutive summer sampling campaigns, its main strength lies in the high-resolution diel and summer temporal sampling of GHG gases, as well several associated biophysical and environmental parameters. However, I recommend moderating the strong extrapolations made from these limited summer datasets to broader seasonal year dynamics. The study would be more robust if it is focused on summer observations and cautiously discussed the potential implications for winter and turnover periods before and after ice cover. I would argue that such extrapolation is quite vague, given that the diel patterns differ between the two sampling days across different years, even in subsequent day measurements in same ecosystem (Figure 5), data varies importantly. Therefore, your statement remains unclear whether these differences are due to actual environmental changes or simply the result of capturing a single day per year, which may not adequately represent diel or yearly variability. Below see other comments about the manuscript that would help you to improve it.

Thank you for this important comment. We agree that the seasonal estimates based on limnophysical indicators and bottom gas concentrations are subject to uncertainty due to the absence of direct year-round GHG flux measurements. In response, we have revised the study's objectives to clarify that the focus of the study is on understanding the drivers of summer temporal variations in GHG fluxes by measuring them, and emphasize the exploration of seasonal patterns using limnophysical proxies, rather than directly measuring seasonal GHG emissions. We also suggest moving the Methodology and corresponding Results sections on seasonal extrapolations, including Table 5, to the Supplementary Material. Consequently, these extrapolations would only be covered in the Discussion section, which we have also modified to improve clarity. Finally, we propose toning down the conclusions regarding seasonal patterns, presenting them now as hypothesis-generating interpretations rather than firm conclusions.

2.    Major concerns

2.1.    Please work on improving the methodology section as it is currently difficult to follow. Consider moving some detailed information to the appendix or supplementary materials, while expanding explanation of key topics. A comprehensive data table would help clarify your sampling approach. For example, it is unclear how many samples were collected for the diel analysis, subsequent days GHG measurements during the summer campaigns, and other parameters. I said this because, I cannot determine the source of

the N values reported in Tables 3 and 4 or whether they come from the diel sampling. Besides, consider consolidating related tables (e.g., Tables 1 and 2 could be merged).

Thank you for your helpful suggestions. Removing the "larger-scale estimates" section (see response to Comment 1) shortened the Methods section by approximately 23%, thereby improving its readability. Regarding the diel analysis, rather than collecting data continuously over a single 24-h period, we measured fluxes and gas concentrations at various times over multiple days. This enabled us to gradually reconstruct a full diel cycle. To clarify the sampling strategy, we propose adding the following to the Methodology:

Line 201: To estimate GHG fluxes, we measured both dissolved gas concentrations ($CO_2$, $CH_4$, and $N_2O$) and $CO_2$ diffusive fluxes using a floating chamber **during the intensive summer campaigns in July 2022 and August 2023. At each lake, measurements were taken over multiple consecutive days across different hours of the day and night. This approach enabled us to reconstruct diel variability in gas fluxes, dissolved gas concentrations, and $k_{600}$, yielding between 9 and 13 data points per lake for both intensive campaigns.**

We also propose presenting the results more transparently and revising the titles of Figure 9 and 10 to improve clarity:

Line 480: **Measurements of $CO_2$ fluxes, dissolved gas concentrations, and $k_{600}$ over multiple consecutive days across different hours allowed us to reconstruct diel variability for both intensive campaigns (Fig. 9).** In 2022, no discernible diel patterns were observed in either the surface concentrations (Fig. S7a,b,c), the gas transfer coefficient (Fig. S8a), or the diffusive emissions of $CO_2$, $CH_4$, and $N_2O$ (Fig. 9a,b,c).

Figure 9. Diel cycles of $CO_2$ (a,d), $CH_4$ (b,e) and $N_2O$ (c,f) diffusive fluxes for the July 2022 (left) and August 2023 (right) intensive summer campaign periods. $CO_2$ fluxes were measured directly with the floating chamber, while $CH_4$ and $N_2O$ fluxes were obtained by applying the k values derived from $CO_2$ fluxes to their respective concentrations. The grey dotted line represents atmospheric equilibrium, and the grey shaded area indicates periods of zero solar radiation. **Although the data are presented on a diurnal cycle, the measurements were taken over a two-week period (see Table S2).** Coloured dotted lines represent sine-fitted trend lines. Time of day is expressed in EST.

To improve the traceability of the samplings, we propose adding Table S4, which will list all dissolved gas concentration measurements used in the study, and modifying Table S6 to include descriptive values for $k_{600}$ rather than just mean, min and max values:

Table S4. Dissolved gases measurements at lakes TAS1 and TAS3 in July 2022 and August 2023. Time of day is expressed in EST.

| Lake | July 2022 | | | | August 2023 | | | |
| | Date | $CO_2$ ($\mu$M) | $CH_4$ ($\mu$M) | $N_2O$ ($\mu$M) | Date | $CO_2$ ($\mu$M) | $CH_4$ ($\mu$M) | $N_2O$ ($\mu$M) |
|---|---|---|---|---|---|---|---|---|
| TAS1 | 2022-07-06 14:20 | 36.30 | 2.26 | 0.015 | 2023-08-09 14:30 | 21.43 | 3.22 | 0.004 |
| | 2022-07-07 08:46 | 40.90 | 1.44 | 0.014 | 2023-08-10 11:50 | 34.07 | 3.13 | 0.004 |
| | 2022-07-07 13:24 | 35.48 | 1.36 | 0.015 | 2023-08-11 18:59 | 24.39 | 1.99 | 0.005 |
| | 2022-07-11 01:46 | 43.34 | 0.92 | 0.015 | 2023-08-12 22:10 | 28.80 | 1.07 | 0.005 |
| | 2022-07-11 16:44 | 29.01 | 0.97 | 0.013 | 2023-08-13 17:59 | 29.19 | 1.42 | 0.004 |
| | 2022-07-12 12:08 | 37.85 | 1.06 | 0.013 | 2023-08-14 10:12 | 46.01 | 1.28 | 0.004 |
| | 2022-07-12 19:08 | 26.96 | 1.06 | 0.013 | 2023-08-14 13:20 | 25.41 | 1.58 | 0.004 |
| | 2022-07-13 09:21 | 40.12 | 0.97 | 0.014 | 2023-08-15 00:40 | 44.05 | 1.20 | 0.004 |
| | 2022-07-13 10:45 | 36.48 | 1.28 | 0.014 | 2023-08-15 16:20 | 34.32 | 1.39 | 0.004 |
| | 2022-07-15 04:33 | 43.20 | 1.57 | 0.013 | 2023-08-16 04:30 | 71.83 | 1.86 | 0.003 |
| | | | | | 2023-08-18 11:00 | 100.14 | 1.41 | 0.003 |
| | | | | | 2023-08-19 7:20 | 122.48 | 2.37 | 0.004 |
| | | | | | 2023-08-20 7:20 | 81.95 | 1.29 | 0.004 |
| TAS3 | 2022-07-06 15:27 | 28.10 | 0.47 | 0.015 | 2023-08-09 15:22 | 28.79 | 1.06 | 0.004 |
| | 2022-07-07 10:24 | 27.95 | 0.59 | 0.015 | 2023-08-11 19:59 | 51.36 | 3.43 | 0.003 |
| | 2022-07-07 14:08 | 24.08 | 0.44 | 0.013 | 2023-08-12 22:59 | 58.84 | 1.87 | 0.003 |
| | 2022-07-08 08:23 | 34.33 | 0.51 | 0.015 | 2023-08-13 19:27 | 30.13 | 1.94 | 0.004 |
| | 2022-07-11 00:24 | 27.53 | 0.45 | 0.014 | 2023-08-14 11:29 | 55.25 | 2.06 | 0.003 |
| | 2022-07-11 16:57 | 16.97 | 0.39 | 0.012 | 2023-08-14 14:22 | 30.57 | 1.82 | 0.003 |
| | 2022-07-12 11:26 | 23.41 | 0.37 | 0.013 | 2023-08-15 01:50 | 218.85 | 3.07 | 0.003 |
| | 2022-07-12 20:07 | 10.15 | 0.30 | 0.012 | 2023-08-15 17:30 | 28.54 | 2.84 | 0.003 |
| | 2022-07-13 08:30 | 32.28 | 0.43 | 0.012 | 2023-08-16 06:59 | 117.43 | 2.69 | 0.003 |
| | | | | | 2023-08-18 12:12 | 154.99 | 5.49 | 0.003 |
| | | | | | 2023-08-19 08:35 | 236.16 | 9.88 | 0.004 |
| | | | | | 2023-08-20 08:20 | 119.01 | 5.50 | 0.004 |

Table S6. Gas transfer velocities ($k_{600}$) measured with the floating chamber in lakes TAS1 and TAS3 using surface GHG concentrations obtained by the headspace method. Time of day is expressed in EST.

| Lake | July 2022 | | August 2023 | |
| | Date | $k_{600}$ (cm h$^{-1}$) | Date | $k_{600}$ (cm h$^{-1}$) |
|---|---|---|---|---|
| TAS1 | 2022-07-07 09:14 | 4.00 | 2023-08-10 11:50 | 10.12 |
| | 2022-07-07 13:36 | 4.18 | 2023-08-11 19:10 | 9.80 |
| | 2022-07-11 01:59 | 4.51 | 2023-08-12 22:41 | 3.19 |
| | 2022-07-11 16:09 | 4.97 | 2023-08-13 18:18 | 4.88 |
| | 2022-07-12 11:53 | 4.08 | 2023-08-14 10:28 | 5.76 |
| | 2022-07-12 19:03 | 2.54 | 2023-08-14 13:48 | 11.01 |
| | 2022-07-13 08:37 | 1.57 | 2023-08-15 01:23 | 1.61 |
| | 2022-07-13 10:51 | 4.29 | 2023-08-15 16:55 | 4.46 |
| | 2022-07-15 04:19 | 1.62 | 2023-08-16 05:09 | 4.37 |
| | | | 2023-08-18 11:41 | 4.13 |
| | | | 2023-08-19 08:00 | 3.09 |
| | | | 2023-08-20 08:00 | 5.36 |
| TAS3 | 2022-07-07 10:41 | 2.89 | 2023-08-11 20:19 | 2.38 |
| | 2022-07-07 14:22 | 1.96 | 2023-08-12 23:36 | 1.60 |
| | 2022-07-08 08:42 | 1.78 | 2023-08-13 19:42 | 3.03 |
| | 2022-07-11 00:59 | 2.57 | 2023-08-14 11:45 | 4.52 |
| | 2022-07-12 11:11 | 1.17 | 2023-08-14 14:42 | 8.19 |
| | 2022-07-12 19:47 | 1.36 | 2023-08-15 02:15 | 0.45 |
| | 2022-07-13 09:50 | 1.48 | 2023-08-15 17:53 | 2.98 |
| | | | 2023-08-16 07:18 | 2.54 |
| | | | 2023-08-18 12:46 | 1.88 |
| | | | 2023-08-19 08:54 | 1.33 |
| | | | 2023-08-20 09:03 | 3.43 |

Regarding Tables 1 and 2, we prefer to keep them separate to maintain clarity on site-specific characteristics. However, Table 4 overlapped with information shown in Figure 7, and therefore we propose moving it to the Supplementary Material.

2.2.  Regarding buoyancy frequency calculations, please clarify how you determined water density in the ponds. Did you incorporate your measurements or use arbitrary values? This section needs a more detailed explanation.

Thank you for this comment. Water density was calculated based on our in-situ measurements of temperature at surface and bottom of the lakes. We have now added this explanation to the methodology section:

Line 164: The buoyancy frequency ($N$, cycles per hour), which quantifies water column stability, was calculated using the following equation:

$$N = \sqrt{-\frac{g}{\rho}\frac{d\rho}{dz}}\frac{3600}{2\pi} \qquad (1)$$

where $g$ is the acceleration due to gravity (9.81 m s$^{-2}$), $\rho$ is the water density (kg m$^{-3}$), and $z$ is the depth (m). **We derived water density from our in situ temperature measurements at the surface and bottom of each lake using the Kell equation (Jones and Harris, 1992). This enabled us to estimate the vertical water density gradient ($d\rho/dz$).**

2.3. The CH$_4$ ebullition methodology requires expansion, particularly since it represented the primary CH$_4$ source to the atmosphere in one pond. Please explain the bubble trap attachment system that prevented movement during the 11-day deployment. Additionally, specify the number of bubble traps used and their spatial distribution across the lake. The issue arises because small replicate ebullition traps may not fully capture ebullition events, Wik et al. (2016, doi: 10.1002/2015GL066501) provided important insights on this topic. Therefore, the argument based on Figure S5 is not valid, the images do not clearly show ebullition, and any observed bubbling could be due to other gases, such as oxygen produced by photosynthesis. If the bubbles were indeed methane, the corresponding diffusive fluxes and surface concentration would be extremely high, which is not supported by your data. Please acknowledge that your dataset is biased, as you mentioned, but avoid overinterpreting the results with vague or unsupported statements.

Thank you for this comment. We agree that the methodology used for measuring CH$_4$ ebullition deserves further clarification, especially given the significance of this flux pathway at TAS3. We have expanded the methodological description in the revised manuscript.

Line 202: Ebullitive fluxes of CO$_2$ and CH$_4$ were measured **at the center of the lake** using an inverted funnel submerged below the water surface.

Line 241: Ebullitive fluxes ($F_E$) of CO$_2$ and CH$_4$ were measured in 2023 following the method outlined by Wik et al. (2013). Bubble traps, consisting of inverted funnels with a collection area of 0.23 m$^2$, were installed for a total 11 days in both lakes (Fig. 2). **The traps were anchored with submerged weights and tied with ropes to fixed shoreline points to minimize movement and drifting. They were positioned near the lake center to improve representativity of ebullition across the central lake area.**

We acknowledge the limitations of using only one bubble trap and the potential for spatial heterogeneity in ebullition rates, as discussed by Wik et al. (2016). However, due to logistical constraints and the small size of the lakes (with diameters of about 10 m), installing multiple traps was not feasible. We now address this limitation in the discussion, emphasizing the need for caution in generalizing ebullitive flux rates from these data and suggesting directions for future work:

Line 605: **However, we acknowledge that our estimates of CH₄ ebullition were based on measurements taken from a single funnel deployed at the center of the lake. This approach restricts the spatial representativeness of the results, especially since ebullition is known to be highly heterogeneous, with localized hotspots often occurring near lake edges or thawing permafrost zones (Wik et al., 2011; Kuhn et al., 2023). This spatial limitation may partially explain the differences observed between the lakes. Future studies should therefore aim to deploy multiple bubble traps across various lake zones to capture this heterogeneity and better constrain whole-lake ebullition estimates.**

We also suggest adding the location of the traps to Figure 1, along with a corresponding note in the caption for clarity:

[Figure]

Figure 1. Location map of the Tasiapik Valley: (a) Map of Canada highlighting the study site marked with a red dot, and (b) areal contours of the two lakes under study. **The locations of the mooring lines are marked with red stars, while the bubble traps are marked with yellow crosses.** Aerial photography by Madeleine St-Cyr.

Regarding Figure S5, we agree that the photographs alone are not sufficient to confirm the presence of CH₄. We have removed Figure S5 and the direct reference to visual evidence of bubbling as support for ebullition. To improve transparency, we have now included the exact bubble content data in the dataset available on Borealis. This information has also been specified in the revised manuscript at line 456.

2.4.   Regarding the CH₄ and N₂O measurements, I was quite surprised by the reported sensitivity of the GC-TCD, especially its ability to detect concentrations close to atmospheric levels for CH₄ and even lower for N₂O. Are you certain that a TCD was used for these analyses? If so, I strongly recommend that you provide

more detailed information on the calibration procedure, the sample injection volume, and the type of column used. This would be highly valuable for researchers working with similar GC-TCD systems. I am particularly skeptical that $N_2O$ concentrations below atmospheric levels can be reliably detected using a TCD. However, if this is indeed the case, please elaborate on the protocols that enabled such sensitivity.

Thank you for your comment. The GC used for our analyses is equipped with 4 detectors (FID X2, TCD and ECD). We used 2 channels into which 5 mL of gas sample is injected and separated in 2 different injection loops. The first, which is fitted with a TCD and FID detector in series, was used to analyse $CH_4$ and $CO_2$. More specifically, the TCD (non-destructive and less sensitive analysis) doses high concentrations, after which the analyte is directed to the FID, which doses lower concentrations. The second, fitted with an ECD detector (which is highly sensitive) analyses nitrous oxide ($N_2O$) by electron capture. We propose the following clarification in the manuscript:

Line 216: Gas samples were analysed using a Trace 1310 gas chromatograph (Thermo Fisher Scientific). **We used two parallel channels, one with TCD-FID detectors (column HSQ 80/100, MS 5A 80/100) for $CH_4$ and $CO_2$, and one with an ECD detector (column HSQ 80/100) for $N_2O$.** Calibration curves were established for $CO_2$ (up to 10 000 ppm), $CH_4$ (up to 45 000 ppm), and $N_2O$ (up to 1 ppm). Detection limits were 200 ppm for $CO_2$, 3 ppm for low $CH_4$ concentrations, 50 ppm for $CH_4$ concentrations above 1000 ppm, and 0.1 ppm for $N_2O$.

**References**

Jones, F. E. and Harris, G. L.: ITS-90 density of water formulation for volumetric standards calibration, J. Res. Natl. Inst. Stand. Technol., 97, 335, 1992.

Kuhn, M. A., Schmidt, M., Heffernan, L., Stührenberg, J., Knorr, K.-H., Estop-Aragonés, C., Broder, T., Gonzalez Moguel, R., Douglas, P. M. J., and Olefeldt, D.: High ebullitive, millennial-aged greenhouse gas emissions from thermokarst expansion of peatland lakes in boreal western Canada, Limnol. Oceanogr., 68, 498-513, 10.1002/lno.12288, 2023.

Wik, M., Crill, P. M., Varner, R. K., and Bastviken, D.: Multiyear measurements of ebullitive methane flux from three subarctic lakes, J. Geophys. Res.-Biogeosciences, 118, 1307-1321, 10.1002/jgrg.20103, 2013.

Wik, M., Crill, P. M., Bastviken, D., Danielsson, Å., and Norbäck, E.: Bubbles trapped in arctic lake ice: Potential implications for methane emissions, J. Geophys. Res.-Biogeosciences, 116, G03044, 10.1029/2011JG001761, 2011.

Wik, M., Varner, R. K., Anthony, K. W., MacIntyre, S., and Bastviken, D.: Climate-sensitive northern lakes and ponds are critical components of methane release, Nat. Geosci., 9, 99-106, 10.1038/Ngeo2578, 2016.